# GENSR: SYMBOLIC REGRESSION BASED ON EQUATION GENERATIVE SPACE

**Qian Li**[1,2]**, Yuxiao Hu**[2,3]**, Juncheng Liu**[4]**, Yuntian Chen**[2*]
[1]Shanghai Jiao Tong University, Shanghai, China
[2]Eastern Institute of Technology, Ningbo, China
[3]The Hong Kong Polytechnic University, Hong Kong, China
[4]Imperial College London, London, England
`qianl01205@sjtu.edu.cn, yuxiao.hu@connect.polyu.hk,`
`junchengliu23@imperial.ac.uk, ychen@eitech.edu.cn`

## ABSTRACT

Symbolic Regression (SR) tries to reveal the hidden equations behind observed data. However, most methods search within a discrete equation space, where the structural modifications of equations rarely align with their numerical behavior, leaving fitting error feedback too noisy to guide exploration. To address this challenge, we propose GenSR[1], a generative latent space–based SR framework following the "map construction $\rightarrow$ coarse localization $\rightarrow$ fine search" paradigm. Specifically, GenSR first pretrains a dual-branch Conditional Variational Autoencoder (CVAE) to reparameterize symbolic equations into a generative latent space with symbolic continuity and local numerical smoothness. This space can be regarded as a well-structured "map" of the equation space, providing directional signals for search. At inference, the CVAE coarsely localizes the input data to promising regions in the latent space. Then, a modified CMA-ES refines the candidate region, leveraging smooth latent gradients. From a Bayesian perspective, GenSR reframes SR task as maximizing the conditional distribution $p(\text{Equ.}|\text{Num.})$, with CVAE training achieving this objective through the Evidence Lower Bound (ELBO). This new perspective provides a theoretical guarantee for the effectiveness of GenSR. Extensive experiments show that GenSR jointly optimizes predictive accuracy, expression simplicity, and computational efficiency, while remaining robust under noise.

## 1 INTRODUCTION

Symbolic Regression (SR) is a core machine learning task aimed at discovering interpretable mathematical expressions that explain observed data. Unlike traditional regression, which assumes a fixed functional form (e.g., linear, polynomial), SR explores the equation space to identify functions that best fit the data. Formally, given $N$ observation samples $\{\boldsymbol{x}_i, y_i\}_{i=1}^{N}$, the objective is to find a mathematical expression $f : \boldsymbol{x} \mapsto y$ that captures the underlying data distribution. Owing to its flexibility and interpretability, SR is widely used in scientific discovery and engineering (Xu et al., 2023; Chen et al., 2022), where understanding the underlying equations is as important as predictive accuracy. SR effectively bridges the gap between black-box models and interpretable scientific insight. Consequently, growing research efforts focus on developing SR algorithms that balance fitting accuracy, computational efficiency, and expression simplicity.

Many prior SR methods search for equations directly in a discrete symbolic space, using heuristic strategies such as genetic programming (GP) (Stephens et al., 2015; Virgolin et al., 2021), Monte Carlo Tree Search (MCTS) (Sun et al.), or other combinatorial optimization methods (Yu et al., 2025; Zhang et al., 2025a). These methods iteratively adjust the symbolic structures of equations through various operations such as crossover, mutation, and tree expansion to search for better solutions. Unfortunately, such modifications do not reliably reduce fitting errors and may even lead the

---

*Corresponding author
[1]`https://github.com/tokaka22/ICLR26-GENSR`

search into suboptimal regions. This is because edit distance and numerical behavior similarity are unrelated measures: equations with similar structures can exhibit drastically different numerical behaviors, and vice versa. Consequently, in the discrete equation space where similarity is measured by edit distance, the numerical error feedback neither corresponds to edit distance nor provides a reliable search direction. The search relies heavily on random mutations, combinations, and backtracking, resulting in an unstable search trajectory and high time complexity. Therefore, the discrete equation space with edit distance is inefficient and even unsuitable for SR.

In this paper, we reparameterize the discrete equation space into a continuous latent space, which can be regarded as a "map" of the equation world (where equations are represented as vectors and dimensions naturally define search directions). To enable efficient search in this space, two key points must be addressed:

(1) **The space should be generative.** A generative latent space, constructed by generative models, supports continuous sampling such that latent vectors decode into syntactically valid equations. This property enables smooth interpolation and fine-grained search in the equation latent space. In contrast, pretraining-based methods (Meidani et al.; Kamienny et al., 2022) typically learn discriminative embeddings for similarity or prediction, rather than modeling a generative distribution. Such discriminative spaces are fragmented and filled with non-decodable regions, hindering continuous search and producing invalid samples.

(2) **The space should be well-organized.** Although some methods attempt to construct a generative latent space for equations (Mežnar et al., 2023; Popov et al., 2023), the Euclidean distances within it reflect only edit distances and are unrelated to the equations' numerical behavior. Consequently, numerical errors cannot be mapped into meaningful Euclidean distances for efficient optimization, and such latent spaces still cannot support smooth, efficient search. Ideally, equations with similar numerical behaviors should be close in the latent space, allowing smooth numerical error signals to guide exploration. Yet expressions like $\cos^2 x$, $1 - \sin^2 x$, and their Taylor expansions exhibit the same numerical behavior but entirely different symbolic forms. Placing such expressions in close proximity would force the decoder to map nearby vectors to drastically different expressions, destabilizing decoding. Thus, a key design challenge is balancing symbolic continuity and numerical smoothness. Inspired by human localization—first identifying a coarse region before precise targeting—we construct a generative latent space with global symbolic continuity and local numerical smoothness. The former ensures stable decoding and induces clear structural clustering for coarse localization, while the latter allows fitting errors to provide directional signals for fine-grained search once the relevant region is located.

Building on this insight, we propose GenSR, a novel framework that follows the "map construction → coarse localization → fine search" paradigm. GenSR first pretrains a dual-branch Conditional Variational Autoencoder (CVAE): The posterior branch learns distributions over symbolic structures given symbolic and numerical inputs, whereas the prior branch learns distributions over numerical features from numerical inputs only. Joint optimization of reconstruction loss and KL divergence yields a **Gen**erative latent space with global symbolic continuity and local numerical smoothness for **SR**, serving as a "map" of the equation world. At inference, input data are mapped to an initial localization distribution by prior branch, which highlights promising latent regions. A modified CMA-ES algorithm then contracts this distribution toward the optimum, exploiting smooth directional signals in the latent space for efficient convergence. From a probabilistic view, GenSR reformulates SR as a Bayesian optimization problem, i.e., maximizing $p(\text{Equ.}|\text{Num.})$, with CVAE training corresponding to ELBO optimization. This perspective provides the theoretical guarantee for GenSR. Extensive experiments show that GenSR consistently achieves an optimal balance of predictive accuracy, expression simplicity, and runtime efficiency, while maintaining robustness under noisy conditions.

Our contributions come in four parts: (1) GenSR constructs a continuous generative latent space unifying symbolic continuity and local numerical smoothness, addressing the inefficiency of discrete equation spaces; (2) GenSR follows a new search paradigm of "map construction → coarse localization → fine search" to achieve effective SR; (3) GenSR provides a new Bayesian perspective for SR and uses ELBO optimization to solve it; (4) Extensive experiments provide a thorough analysis of latent space properties and demonstrate that our method achieves a better balance among accuracy, equation complexity, and efficiency.

## 2 RELATED WORK

The field of SR has evolved through a variety of approaches. The most common SR methods rely on combinatorial optimization over a discrete equation space using heuristic search. Genetic Programming (GP) techniques (Stephens et al., 2015; Virgolin et al., 2021; Burlacu et al., 2020; Cranmer, 2023) evolve populations of expressions through selection and mutation, while Monte Carlo Tree Search (MCTS) methods (Sun et al.) systematically explore the expression tree space. More recent neural-guided approaches have integrated reinforcement learning (Petersen et al.), Transformer-based planning (Shojaee et al., 2023), retrieval-augmented generation (Zhang et al., 2025a), and minimum description length principles (Yu et al., 2025) to enhance search efficiency. Large language models have also been applied to generate equations through prompting techniques (Shojaee et al.; Grayeli et al., 2024); however, due to inherent hallucination issues, LLM-based methods still resemble enumeration in the discrete space and require many queries to reach high-quality solutions. Ultimately, symbolic regression over discrete equation spaces remains fundamentally inefficient, as structural similarity poorly reflects numerical behavior, leaving search unguided and error feedback uninformative.

Alternative strategies have sought to avoid discrete search entirely. Sequence-to-sequence models (Kamienny et al., 2022; Li et al., 2022; Biggio et al., 2021) generate equations directly but do not enforce alignment between symbolic forms and numerical behavior during training. limiting their generalization. The SNIP framework (Meidani et al.) made progress by learning a shared latent space through contrastive learning, creating representations useful for similarity assessment. However, because SNIP learns a discriminative rather than generative space, many points in its latent space do not correspond to valid equations, making it inadequate for the generative search demands of SR.

Our work addresses these limitations by learning a continuous, generative latent space using a dual-branch CVAE. This approach transforms the discrete equation space into a structured manifold where nearby points correspond to expressions with similar symbolic forms and numerical properties. The resulting space provides meaningful gradients that enable an efficient three-stage search strategy—map construction, coarse localization, and fine-grained search—directly addressing the core weaknesses of previous methods.

## 3 GENSR: SYMBOLIC REGRESSION BASED ON **Gen**ERATIVE LATENT SPACE

As illustrated in Fig. 1, the core idea of GenSR is to construct a continuous generative latent space for equations (which can be regarded as a "map" of the equation world) using a dual-branch CVAE and to perform efficient search directly within this space. The dual-branch CVAE consists of a Transformer-based encoder–decoder network, two branch-specific networks, and a feature fusion module. The CVAE is pre-trained on approximately 5 million synthetic equation–sample pairs to ensure coverage of diverse functional forms. At inference, GenSR leverages the prior branch to produce an initial localization distribution in the latent space and refines the solution in the latent space using a degenerate version of CMA-ES (Hansen, 2016), enabling fast and precise search for the distribution of candidate equations.

### 3.1 PREPARATIONS

To construct the generative latent space, we first prepare synthetic data and preprocess input numerical samples and equations, laying the foundation for dual-branch CVAE pre-training.

**Preparation of synthetic data.** To construct the generative latent space, we pretrain the conditional VAE (Fig. 1) on a large synthetic dataset. Following Kamienny et al. (2022); Meidani et al., we construct a dataset in which each equation $f$ is paired with $m$ numerical examples $(\boldsymbol{x}, y)_{i=1}^{m} \in \mathbb{R}^{m \times (D+1)}$, where $D$ denotes the dimensionality of the input variables. Note that we control the expression length to push complex equations toward the periphery of the latent space, mitigating overfitting to complex solutions during search.

**Pre-processing of numeric samples.** Following Kamienny et al. (2022), we tokenize each numerical sample using base-10 floating-point representation, rounding to four significant digits and decomposing into sign, mantissa (0–9999), and exponent ($E{-}100$–$E100$), e.g., 0.7895 is tok-

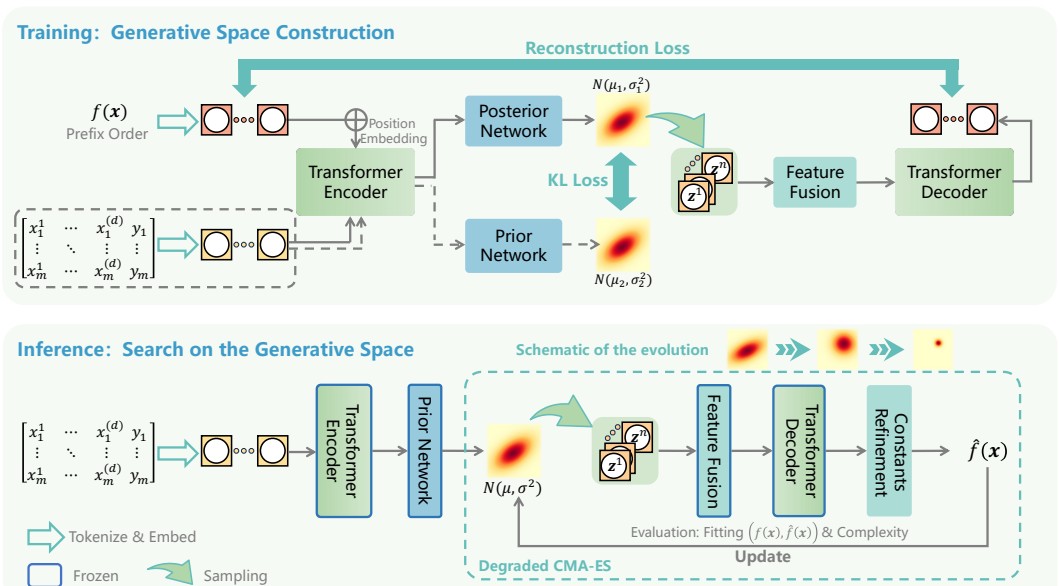

Figure 1: The overview of GenSR. During training, the dashed lines denote the prior branch, while the solid lines indicate the posterior branch. During inference, only the prior branch is used.

enized as $[+, 7895, E-4]$. The $m$ samples for each equation $f$ are represented as $\{(\boldsymbol{x}, y)\}_{i=1}^{m} \in \mathbb{R}^{m \times 3(D+1)}$. A learnable embedder with dimensionality reduction maps these token embeddings to $X \in \mathbb{R}^{m \times d}$, which serves as the numerical input to the Transformer encoder (Kamienny et al., 2022).

**Pre-processing of equation.** Each expression $f$ is represented as a binary tree and linearized in prefix order (Lample & Charton). Operators, variables, and integers are mapped to dedicated tokens, while constants are tokenized in the same way as numeric samples. Special tokens $[\langle\text{BOS}\rangle]$ and $[\langle\text{EOS}\rangle]$ are inserted to mark the start and end of each sequence. Particularly, sequences are padded to a fixed length $m$ to match the sequence length of numeric samples. Each token is embedded into a $d$-dimensional space with positional encoding, yielding $\boldsymbol{F} \in \mathbb{R}^{m \times d}$ as the equation input to the Transformer encoder.

## 3.2 PRE-TRAIN THE GENERATIVE LATENT SPACE FOR EQUATIONS

We adopt the dual-branch CVAE shown in Fig. 1 to learn a well-organized generative latent space for equations, which can be regarded as a "map" of the equation space. This map provides initial localization of high-probability regions and enables smooth directional search toward optimal solutions, achieving efficient equation discovery. The dual-branch CVAE consists of a posterior branch and a prior branch. Following Kingma & Welling (2013); Sohn et al. (2015), we assume the conditional distribution of latent variables is a multivariate independent Gaussian distribution. Next, we describe each branch and its training details.

**Posterior Branch** (solid lines in Fig. 1): Comprises an 8-layer Transformer encoder, a posterior network, a feature fusion MLP, and an 8-layer Transformer decoder. It takes both $X$ (numerical samples) and $F$ (symbolic equation) as inputs, learning the posterior distribution $q(\boldsymbol{z}|\boldsymbol{X}, \boldsymbol{F}) = \mathcal{N}(\boldsymbol{\mu}_1, \boldsymbol{\sigma}_1^2\boldsymbol{I})$ via the posterior network. Using the reparameterization trick, we sample $n$ latent vectors $\boldsymbol{Z} = \{\boldsymbol{z}_i = \boldsymbol{\mu}_1 + \boldsymbol{\sigma}_1 \odot \boldsymbol{\epsilon}\}_{i=1}^{n}$, where $\boldsymbol{\epsilon} \sim \mathcal{N}(0, I)$ and $\odot$ denotes element-wise multiplication. The feature fusion MLP aggregates salient information across sampled latent vectors, and the Transformer decoder reconstructs the original symbolic equation, ensuring the latent space captures the continuity of symbolic structures.

**Prior Branch** (dashed lines in Fig. 1): The prior branch shares all modules (Transformer encoder/decoder, feature fusion MLP) with the posterior branch, except that the posterior network is replaced by a prior network. It takes only numerical samples $X$ as input and learns the prior distribution $p(\boldsymbol{z}|\boldsymbol{X}) = \mathcal{N}(\boldsymbol{\mu}_2, \boldsymbol{\sigma}_2^2\boldsymbol{I})$, which captures the numerical behavior of equations.

**Training Process**: During training, both branches are jointly optimized on paired data $(\boldsymbol{F}, \boldsymbol{X})$ using a combined objective: (1) Reconstruction loss $L_{\text{rec}}$ enforces continuity of symbolic structures by accurately decoding equations from the latent space; (2) KL divergence $D_{\text{KL}}$ aligns the posterior distribution $q(\boldsymbol{z}|\boldsymbol{X}, \boldsymbol{F})$ with the prior distribution $p(\boldsymbol{z}|\boldsymbol{X})$, ensuring local numerical smoothness. This alignment of distributions regularizes the prior network to support reliable inference when the true equation is unknown. We apply KL annealing during training to mitigate posterior collapse.

## 3.3 ANALYSIS OF GENERATIVE LATENT SPACE PROPERTIES

We further explain why GenSR in Sec. 3.2 yields a generative latent space that is globally continuous in symbolic structure and locally smooth in numerical features.

**Continuity of symbolic structure.** Let $q^{(1)}(\boldsymbol{z}) = \mathcal{N}(\boldsymbol{\mu}^{(1)}, \boldsymbol{\sigma}^{(1)2}\boldsymbol{I})$ and $q^{(2)}(\boldsymbol{z}) = \mathcal{N}(\boldsymbol{\mu}^{(2)}, \boldsymbol{\sigma}^{(2)2}\boldsymbol{I})$ denote the two output posterior distributions for input sample pairs $(\boldsymbol{F}^{(1)}, \boldsymbol{X}^{(1)})$ and $(\boldsymbol{F}^{(2)}, \boldsymbol{X}^{(2)})$, respectively. We define the high-confidence region for each distribution at confidence level $\alpha$ as:

$$\mathcal{R}_\alpha^{(i)} = \left\{ \boldsymbol{z} \in \mathbb{R}^d : \sum_{j=1}^d \frac{(z_j - \mu_j^{(i)})^2}{(\sigma_j^{(i)})^2} \leq \chi_{d,\alpha}^2 \right\}, \quad i = 1, 2,$$

where $\chi_{d,\alpha}^2$ is the $\alpha$-quantile of the chi-squared distribution with $d$ degrees of freedom. For two distributions, their $n$-sampled sets $\boldsymbol{Z}^{(j)} = \{\boldsymbol{z}_i^{(j)} = \boldsymbol{\mu}^{(j)} + \boldsymbol{\sigma}^{(j)} \odot \boldsymbol{\epsilon}\}_{i=1}^n$, where $\boldsymbol{\epsilon} \sim \mathcal{N}(0, \boldsymbol{I})$, $j = 1, 2$, typically fall within the respective high-probability regions $\mathcal{R}_\alpha^{(1)}$ and $\mathcal{R}_\alpha^{(2)}$. Here, unlike the traditional VAE framework that samples only a single $\boldsymbol{z}$, we employ repeated probabilistic sampling and a feature fusion module to ensure that the reconstruction loss influences the high-probability regions of the distribution, thereby strengthening the symbolic continuity of the latent space. If there exists $\boldsymbol{z}^* \in \mathcal{R}_\alpha^{(1)} \cap \mathcal{R}_\alpha^{(2)}$, it is shaped by the reconstruction objectives of both $f^{(1)}$ and $f^{(2)}$, yielding a decoded structure that interpolates between them. This encourages a smooth symbolic transition in the latent space. Trained on diverse equations, such local interpolation generalizes globally, clustering structurally similar equations in adjacent regions and inducing a latent space with continuous symbolic structure.

**Smoothness of numerical feature.** We leverage the prior branch to capture characteristics of numerical distributions and align them with the symbolic latent space through a KL divergence loss, imposing numerical constraints on the distribution. This design resembles contrastive learning in SNIP (Meidani et al.), but with a key distinction: we align entire distributions rather than individual points, which ensures local numerical smoothness within the generative latent space. Specifically, the KL loss $D_{\text{KL}}(q(\boldsymbol{z}|\boldsymbol{X}, \boldsymbol{F}) \parallel p(\boldsymbol{z}|\boldsymbol{X}))$ regularizes the posterior distribution to stay close to the prior distribution. For the independent Gaussian distributions assumed in our CVAE ($\boldsymbol{\Sigma}_1 = \boldsymbol{\sigma}_1^2\boldsymbol{I}$, $\boldsymbol{\Sigma}_2 = \boldsymbol{\sigma}_2^2\boldsymbol{I}$), this divergence simplifies to:

$$D_{\text{KL}}(q \parallel p) = \frac{1}{2} \sum_{j=1}^d \left( \frac{(\mu_{1,j} - \mu_{2,j})^2}{\sigma_{2,j}^2} + \frac{\sigma_{1,j}^2}{\sigma_{2,j}^2} - \ln \frac{\sigma_{1,j}^2}{\sigma_{2,j}^2} - 1 \right).$$

Minimizing this term in our dual-branch CVAE constrains the distance between the posterior and prior means while aligning their variances, ensuring that the high-probability regions—rather than isolated points—are matched to preserve local numerical smoothness.

Consequently, GenSR establishes a well-structured generative latent space for SR. We will provide more validation of the properties of the space in Sec. 5.3.

## 3.4 SEARCH IN THE GENERATIVE LATENT SPACE

During inference, since the ground-truth equation is unavailable, only the prior branch is used. The inference process is illustrated in Fig. 1.

For a given set of numerical samples $\{(\boldsymbol{x}, y)_i\}_{i=1}^m$, we first preprocess them as described in Sec. 3.1. The prior branch then encodes them into an initial Gaussian distribution $\mathcal{N}(\boldsymbol{\mu}_0, \boldsymbol{\sigma}_0^2\boldsymbol{I})$ in the latent space. This distribution provides coarse localization, highlighting regions likely to contain equations

that fit the data well. For example, high-probability regions align with plausible symbolic families (e.g., trigonometric, exponential) and input dimensionality, offering a structured starting point for fine-grained search.

Since the prior and posterior latent spaces are only approximate, directly decoding the vector $\boldsymbol{\mu}_0$ in an end-to-end manner may not yield the most accurate result. To refine the initial distribution $\mathcal{N}(\boldsymbol{\mu}_0, \boldsymbol{\sigma}_0^2 \boldsymbol{I})$ toward the optimal solution, we use a modified Covariance Matrix Adaptation Evolution Strategy (CMA-ES) (Hansen, 2016), leveraging the numerical smoothness of the latent space for efficient convergence. The algorithm is as follows:

1. **Initialization**: Set the initial search distribution as $\mathcal{N}(\boldsymbol{\mu}_0, \boldsymbol{\sigma}_0^2 \boldsymbol{I})$ (with $\boldsymbol{\mu}_0$ and $\boldsymbol{\sigma}_0$ are the output of the prior network).

2. **Iteration** (for each generation $i$):
   (a) Sample multiple latent vectors $\{\boldsymbol{z}_j\}$ from $\mathcal{N}(\boldsymbol{\mu}_i, \boldsymbol{\sigma}_i^2 \boldsymbol{I})$.
   (b) Decode $\{\boldsymbol{z}_j\}$ into candidate equations $\{f_j(\boldsymbol{x})\}$ using the well-trained decoder.
   (c) Refine the constants of $\{f_j(\boldsymbol{x})\}$ via BFGS optimization.
   (d) Evaluate the fitness of candidate equations: Fitness $= R^2 - \omega \cdot$ complexity.
   (e) Select the top-$p$ candidates and update the distribution parameters $\boldsymbol{\mu}_i \to \boldsymbol{\mu}_{i+1}$ and $\boldsymbol{\sigma}_i \to \boldsymbol{\sigma}_{i+1}$ using the CMA-ES update rules. This yields the updated distribution

The prior branch provides a well-organized generative latent space for CMA-ES, thereby enabling faster and more stable convergence. However, CMA-ES becomes costly as the latent dimension $d$ grows. To improve efficiency, we introduce two key modifications to standard CMA-ES:

1. **Diagonal Covariance Assumption**: Inspired by Ros & Hansen (2008), we restrict the covariance matrix $\boldsymbol{\Sigma}$ to a diagonal form $\boldsymbol{\sigma}^2 \boldsymbol{I}$, updating only the variance of each latent dimension independently. This aligns with the latent variable assumption in VAEs (Kingma & Welling, 2013) (modeled as independent Gaussian components) and reduces the number of parameters to optimize.

2. **Top-$k$ Variance Update**: Update only the top $k$ latent dimensions with the largest variances (while setting others to zero). This focuses search on the most uncertain and relevant directions, accelerating convergence without compromising solution quality.

These modifications enable rapid and precise contraction of the initial distribution, facilitating rapid localization of the target equation. Fig. 1 provides a schematic of the CMA-ES distribution contraction process.

## 4 GENSR: SYMBOLIC REGRESSION FROM A BAYESIAN PERSPECTIVE

GenSR can be naturally interpreted as probabilistic inference, reformulating SR task as a Bayesian problem: given numerical samples $\boldsymbol{X} \in \mathbb{R}^{m \times d}$, the goal is to infer a posterior distribution $p(\boldsymbol{F}|\boldsymbol{X})$ over equation space that best explains the data, i.e., maximize $p(\boldsymbol{F}|\boldsymbol{X})$. Unlike traditional SR tasks seeking a single optimum, this Bayesian formulation reasons over the entire posterior distribution of candidate equations, enabling principled uncertainty quantification and probabilistic comparison across candidates.

To approximate the intractable distribution $p(\boldsymbol{F}|\boldsymbol{X})$, we introduce a variational distribution $q(\boldsymbol{z}|\boldsymbol{X}, \boldsymbol{F})$, where $\boldsymbol{z} \in \mathbb{R}^d$ is a latent vector. Following Sohn et al. (2015), we derive the Evidence Lower Bound (ELBO) of the marginal log-likelihood $\log p(\boldsymbol{F}|\boldsymbol{X})$ as follows. The full derivation is provided in Appendix B.

$$
\begin{aligned}
\log p(\boldsymbol{F}|\boldsymbol{X}) &= \int_z q(\boldsymbol{z}|\boldsymbol{X}, \boldsymbol{F}) \log p(\boldsymbol{F}|\boldsymbol{X}) d\boldsymbol{z} \\
&\geq \int_z q(\boldsymbol{z}|\boldsymbol{X}, \boldsymbol{F}) \log \frac{p(\boldsymbol{F}, \boldsymbol{z}|\boldsymbol{X})}{q(\boldsymbol{z}|\boldsymbol{X}, \boldsymbol{F})} d\boldsymbol{z} \\
&= \mathbb{E}_{q(\boldsymbol{z}|\boldsymbol{X}, \boldsymbol{F})}\left[\log p(\boldsymbol{F}|\boldsymbol{X}, \boldsymbol{z})\right] - D_{\mathrm{KL}}\left(q(\boldsymbol{z}|\boldsymbol{X}, \boldsymbol{F}) \,\|\, p(\boldsymbol{z}|\boldsymbol{X})\right).
\end{aligned} \tag{1}
$$

Maximizing $p(\boldsymbol{F}|\boldsymbol{X})$ amounts to maximizing the ELBO 1. To optimize this objective, GenSR employs posterior encoder $q_{\phi,\varphi}(\boldsymbol{z}|\boldsymbol{X}, \boldsymbol{F})$, prior encoder $p_{\phi,\psi}(\boldsymbol{z}|\boldsymbol{X})$, and decoder $p_\theta(\boldsymbol{F}|\boldsymbol{X}, \boldsymbol{z})$ to parameterize $q(\boldsymbol{z}|\boldsymbol{X}, \boldsymbol{F})$, $p(\boldsymbol{z}|\boldsymbol{X})$, and $p(\boldsymbol{F}|\boldsymbol{X}, \boldsymbol{z})$, respectively. Then, training our dual-branch CVAE with $L_{rec}$ and $D_{KL}$ is equivalent to maximizing the ELBO 1: maximizing

$\mathbb{E}_{q(\boldsymbol{z}|\boldsymbol{X},\boldsymbol{F})}[\log p(\boldsymbol{F}|\boldsymbol{X},\boldsymbol{z})]$ corresponds to maximizing the probability of reconstructing $\boldsymbol{F}$ using $\boldsymbol{z}$ sampled from the output of $q_{\phi,\varphi}(\boldsymbol{z}|\boldsymbol{X},F)$, i.e., minimizing the reconstruction loss $L_{rec}$; and the second term in inequality 1 corresponds to the KL-divergence loss in Sec. 3.2 which aligns the variational distribution $q_{\phi,\varphi}(\boldsymbol{z}|\boldsymbol{X},\boldsymbol{F})$ with the prior distribution $p_{\phi,\psi}(\boldsymbol{z}|\boldsymbol{X})$. During inference, we sample $\boldsymbol{z}$ from the prior distribution $p_{\phi,\psi}(\boldsymbol{z}|\boldsymbol{X})$ to reconstruct the equation $\boldsymbol{F}$, which approximates the intractable expectation $\mathbb{E}_{q_{\phi}(\boldsymbol{z}|\boldsymbol{X},\boldsymbol{F})}[\log p_{\theta}(\boldsymbol{F}|\boldsymbol{X},\boldsymbol{z})]$ with $\mathbb{E}_{p_{\psi}(\boldsymbol{z}|\boldsymbol{X})}[\log p_{\theta}(\boldsymbol{F}|\boldsymbol{X},\boldsymbol{z})]$. To mitigate the inherent approximation error, we refine the prior distribution as explained in Sec. 3.4.

Notably, GenSR differs from Holt et al., which approximates $p(\text{Token}|\text{Num.})$ at the token level without constructing a latent distribution over symbolic equations, and also from Mežnar et al. (2023); Popov et al. (2023), which only models $p(\text{Equ.})$. GenSR explicitly formulates the SR task as maximizing the conditional distribution $p(\text{Equ.}|\text{Num.})$ from a Bayesian inference perspective. To the best of our knowledge, GenSR is the first framework that realize SR based on estimating and optimizing the ELBO of $p(\text{Equ.}|\text{Num.})$. Under GenSR framework, our dual-branch CVAE pre-training model corresponds directly to a ELBO, while the CMA-ES refinement process serves as an approximate optimization of the conditional variational distribution. This not only provides GenSR with solid theoretical guarantee, but also introduces a new Bayesian perspective and technical route for SR.

# 5 EXPERIMENT

## 5.1 EXPERIMENTAL SETTINGS

To validate our method, we conduct evaluations on the SRBench benchmark (La Cava et al., 2021; Cavalab, 2022), which consists of 119 Feynman equations, 14 ODE-Strogatz challenges, and 57 black-box regression tasks without known underlying functions. We compare against 18 baseline algorithms, covering both pretraining-based methods and diverse heuristic-based methods. The evaluation metrics include accuracy $R^2$, time complexity, and equation complexity. To better balance and visualize the trade-offs among these three metrics, we employ the Pareto front for representation. Comprehensive information on datasets, metrics, and baseline methods can be found in Appendix G.1, G.2, and G.3. For clarity, this section presents only the key experimental results; the full results are deferred to Appendix G.5. Additional analyses, including the ablation study and other task based on our generative space, are provided in Appendix E, G.4.

## 5.2 COMPARISON OF GENSR WITH OTHER METHODS

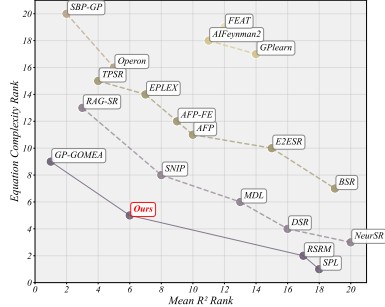 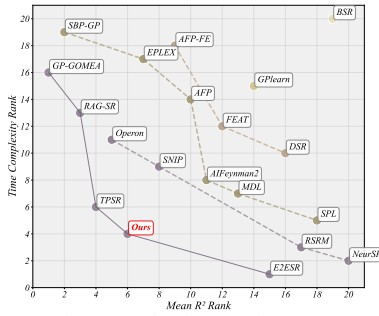

Figure 2: Pareto front results on the Feynman dataset. The $x$-axis shows the mean test $R^2$ rank, while the $y$-axis shows equation complexity rank (left) and time complexity rank (right). Solid lines indicate the optimal Pareto front, and dashed lines show lower-ranked fronts from bottom-left to top-right.

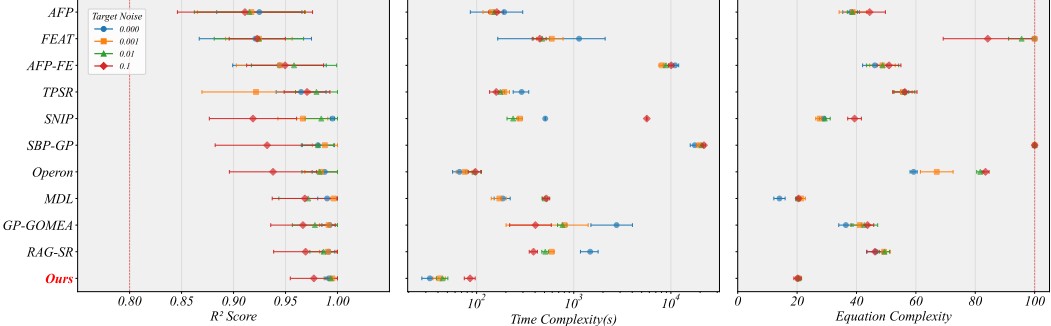

Figure 3: Comparison on the Strogatz dataset under different noise levels. Subplots (left to right) report $R^2$ score, time complexity (s), and equation complexity. Noise levels are represented by blue circles (0.000), orange squares (0.001), green triangles (0.01), and red diamonds (0.1), with error bars indicating standard deviations. Only methods whose mean $R^2$ across noise settings exceeds 0.9 are included.

**Pareto performance analysis.** As shown in Fig. 2, our method achieves a balanced optimization across accuracy ($R^2$), equation complexity, and time complexity, and consistently lies on the rank-1 Pareto front, outperforming competing baselines overall. The superiority stems from the generative continuous latent space learned via CVAE training, which preserves global structural continuity while ensuring local numerical smoothness. Built upon this representation, the GenSR efficiently enables a hierarchical optimization paradigm of "map construction → coarse localization → fine search." In contrast to discrete symbolic spaces that rely on heuristic mutations and backtracking, our well-organized generative latent space provides stable directional signals, making the search both more efficient and less prone to overfitting. Consequently, GenSR achieves robust improvements in accuracy, expression brevity, and computational efficiency.

**Performance under noisy targets.** As shown in Fig. 3, our method achieves the highest $R^2$ across varying noise levels, with smaller variance than competing baselines, indicating stable and robust fitting under perturbations. Moreover, it requires significantly less runtime and yields equations of lower complexity. These results highlight that our generative latent space preserves structural organization and local smoothness even in the presence of noise, thereby providing search with stable optimization directions and avoiding the random oscillations and inefficient exploration typical of discrete symbolic spaces. Benefiting from this design, GenSR maintains a favorable balance of accuracy, efficiency, and compactness under noisy conditions, demonstrating strong robustness.

## 5.3 VISUALIZATION COMPARISON OF LATENT SPACE

We compare GenSR with E2ESR (Kamienny et al., 2022) and SNIP (Meidani et al.). E2ESR employs a Transformer to predict target equations, while SNIP leverages contrastive pretraining to align symbolic forms with numeric data. Unlike GenSR, Etheir spaces are all discriminative rather than generative.

**Structural separability comparison of latent spaces.** As shown in Fig. 4, different methods induce distinct distributional characteristics of equation latent vectors. The latent space of E2ESR fails to effectively separate function types and input dimensions: although it can partially distinguish two input variants of the log operator, the trig and exp components remain highly entangled, resulting in a mixed representation. SNIP achieves slightly better separation at the operator level, yet lacks intra-class compactness; for instance, the exp-5D cluster splits into two distinct regions, accompanied by noticeable overlaps across categories. In contrast, our GenSR exhibits a clearly structural disentangled latent space, where both different function types and input dimensionality are well separated. This space provides GenSR with favorable conditions for initial equation localization while ensuring decoding stability.

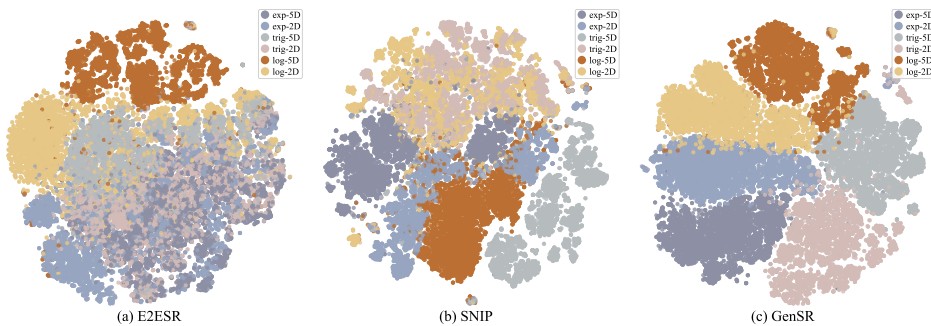

Figure 4: 2D t-SNE visualization of latent variables from E2ESR, SNIP, and GenSR. The legend distinguishes six categories, corresponding to equations from three representative function families, each evaluated under 2D and 5D input dimensionality, illustrating the clustering behavior of the learned latent spaces.

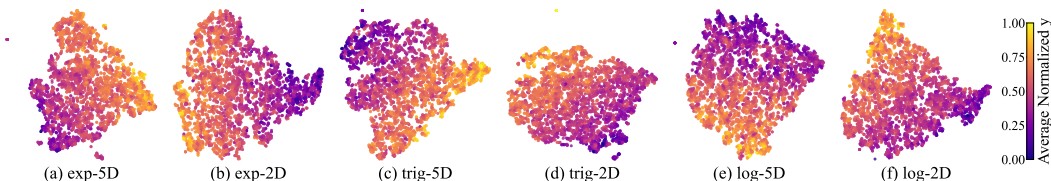

Figure 5: 2D t-SNE visualization of GenSR latent variables for equations from three function families (exponential, trigonometric, logarithmic) under 2D and 5D input settings, shown in subplots (a)–(f). Colors indicate the average of normalized $y$ values, as displayed in the accompanying color bar.

**Numerical continuity within function Classes.** As for each observed sample $\{\boldsymbol{x}_i, y_i\}_{i=1}^N$, we use the normalized mean of $\{y_i\}_{i=1}^N$ to represent the numerical feature. Fig. 5 visualizes the local numerical feature distribution of GenSR, where each subfigure shows the numerical feature distribution of one cluster in Fig. 4 (c). For each structural cluster, GenSR exhibits a smooth distribution of numerical features, which enables the search algorithm refine convergence along smooth numerical directions after localizing high probability areas. Details and additional latent space experiments are provided in the Appendix C.

## 6 CONCLUSION AND FUTURE WORK

We introduced GenSR, a generative latent space-based SR framework that maps the discrete symbolic equation space into a continuous, well-structured space. GenSR first jointly enforces symbolic continuity and local numerical smoothness through a dual-branch CVAE, and then enables efficient symbolic regression through coarse localization and fine-grained search. This design bridges combinatorial symbolic search with continuous optimization, yielding a balanced trade-off among accuracy, complexity, and efficiency. Moreover, GenSR organizes distinct equation families into separate latent regions, naturally supporting tasks such as equation classification, which we further demonstrate in the appendix. These results highlight the broader potential of generative latent spaces as a foundation for equation discovery and, more generally, for combinatorial scientific search. Future works include incorporating richer generative priors and physical constraints, and leveraging more advanced generative models to construct more powerful and generalizable equation spaces, thereby advancing interpretable and domain-aware machine learning for scientific discovery.

## 7 ACKNOWLEDGEMENT

This work was supported by the National Natural Science Foundation of China (12572266), and the High Performance Computing Centers at Eastern Institute of Technology, Ningbo, and Ningbo Institute of Digital Twin.

## REPRODUCIBILITY STATEMENT

For the reproducibility of our results, we have provided a detailed description of our methods and experimental setups in Sec. 3.1, 3.2 and Appendix F. We also confirmed the robustness of our results through the experiment (Appendix G.5). In addition, to further facilitate the reproduction, we will release our codes and the checkpoints for the trained models.

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

## A    THE USE OF LARGE LANGUAGE MODELS (LLMs)

Large Language Models (LLMs) were used solely as auxiliary tools for language polishing and minor editing. They did not contribute to the research ideation, experimental design, or writing to an extent that would qualify them as contributors. All substantive research ideas, methodologies, and results presented in this paper are entirely the work of the authors.

## B    DERIVATION OF $p(\boldsymbol{F}|\boldsymbol{X})$MMXIMIZATION

We refer to Sohn et al. (2015) to approximate the intractable distribution $p(\boldsymbol{F}|\boldsymbol{X})$ using a variational distribution $q(\boldsymbol{z}|\boldsymbol{X}, \boldsymbol{F})$ that can be any distribution (which is assumed as a multivariate independent Gaussian distribution in our work). The detailed derivation is as follows.

$$
\begin{aligned}
\log p(\boldsymbol{F}|\boldsymbol{X}) &= \int_{\boldsymbol{z}} q(\boldsymbol{z}|\boldsymbol{X}, \boldsymbol{F}) \log p(\boldsymbol{F}|\boldsymbol{X}) d\boldsymbol{z} \\
&= \int_{\boldsymbol{z}} q(\boldsymbol{z}|\boldsymbol{X}, \boldsymbol{F}) \log \frac{p(\boldsymbol{F}, \boldsymbol{z}|\boldsymbol{X})}{p(\boldsymbol{z}|\boldsymbol{X}, \boldsymbol{F})} d\boldsymbol{z} \\
&= \int_{\boldsymbol{z}} q(\boldsymbol{z}|\boldsymbol{X}, \boldsymbol{F}) \log \left( \frac{p(\boldsymbol{F}, \boldsymbol{z}|\boldsymbol{X})}{q(\boldsymbol{z}|\boldsymbol{X}, \boldsymbol{F})} \frac{q(\boldsymbol{z}|\boldsymbol{X}, \boldsymbol{F})}{p(\boldsymbol{z}|\boldsymbol{F}, \boldsymbol{X})} \right) d\boldsymbol{z} \\
&= \int_{\boldsymbol{z}} q(\boldsymbol{z}|\boldsymbol{X}, \boldsymbol{F}) \log \left( \frac{p(\boldsymbol{F}, \boldsymbol{z}|\boldsymbol{X})}{q(\boldsymbol{z}|\boldsymbol{X}, \boldsymbol{F})} \right) d\boldsymbol{z} + \int_{\boldsymbol{z}} q(\boldsymbol{z}|\boldsymbol{X}, \boldsymbol{F}) \log \left( \frac{q(\boldsymbol{z}|\boldsymbol{X}, \boldsymbol{F})}{p(\boldsymbol{z}|\boldsymbol{F}, \boldsymbol{X})} \right) d\boldsymbol{z} \\
&= \int_{\boldsymbol{z}} q(\boldsymbol{z}|\boldsymbol{X}, \boldsymbol{F}) \log \left( \frac{p(\boldsymbol{F}, \boldsymbol{z}|\boldsymbol{X})}{q(\boldsymbol{z}|\boldsymbol{X}, \boldsymbol{F})} \right) d\boldsymbol{z} + D_{KL} \left( q(\boldsymbol{z}|\boldsymbol{X}, \boldsymbol{F}) \| p(\boldsymbol{z}|\boldsymbol{F}, \boldsymbol{X}) \right).
\end{aligned}
$$

Since $D_{KL} \left( q(\boldsymbol{z}|\boldsymbol{X}, \boldsymbol{F}) \| p(\boldsymbol{z}|\boldsymbol{F}, \boldsymbol{X}) \right) \geq 0$, thus, we have the lower bound of $\log p(\boldsymbol{F}|\boldsymbol{X})$:

$$
\begin{aligned}
\log p(\boldsymbol{F}|\boldsymbol{X}) &\geq \int_{\boldsymbol{z}} q(\boldsymbol{z}|\boldsymbol{X}, \boldsymbol{F}) \log \left( \frac{p(\boldsymbol{F}, \boldsymbol{z}|\boldsymbol{X})}{q(\boldsymbol{z}|\boldsymbol{X}, \boldsymbol{F})} \right) d\boldsymbol{z} \\
&= \int_{\boldsymbol{z}} q(\boldsymbol{z}|\boldsymbol{X}, \boldsymbol{F}) \log \left( \frac{p(\boldsymbol{F}|\boldsymbol{z}, \boldsymbol{X}) p(\boldsymbol{z}|\boldsymbol{X})}{q(\boldsymbol{z}|\boldsymbol{X}, \boldsymbol{F})} \right) d\boldsymbol{z} \\
&= \int_{\boldsymbol{z}} q(\boldsymbol{z}|\boldsymbol{X}, \boldsymbol{F}) \log p(\boldsymbol{F}|\boldsymbol{z}, \boldsymbol{X}) d\boldsymbol{z} + \int_{\boldsymbol{z}} q(\boldsymbol{z}|\boldsymbol{X}, \boldsymbol{F}) \log \left( \frac{p(\boldsymbol{z}|\boldsymbol{X})}{q(\boldsymbol{z}|\boldsymbol{X}, \boldsymbol{F})} \right) d\boldsymbol{z} \\
&= \int_{\boldsymbol{z}} q(\boldsymbol{z}|\boldsymbol{X}, \boldsymbol{F}) \log p(\boldsymbol{F}|\boldsymbol{z}, \boldsymbol{X}) d\boldsymbol{z} - D_{KL} \left( q(\boldsymbol{z}|\boldsymbol{X}, \boldsymbol{F}) \| p(\boldsymbol{z}|\boldsymbol{X}) \right) \\
&= \mathbb{E}_{q(\boldsymbol{z}|\boldsymbol{X}, \boldsymbol{F})} \left[ \log p(\boldsymbol{F}|\boldsymbol{X}, \boldsymbol{z}) \right] - D_{KL} \left( q(\boldsymbol{z}|\boldsymbol{X}, \boldsymbol{F}) \| p(\boldsymbol{z}|\boldsymbol{X}) \right).
\end{aligned}
$$

To optimize this lower bound, GenSR employ posterior encoder $q_{\phi,\varphi}(\boldsymbol{z}|\boldsymbol{X}, \boldsymbol{F})$, prior encoder $p_{\phi,\psi}(\boldsymbol{z}|\boldsymbol{X})$, and decoder $p_{\theta}(\boldsymbol{F}|\boldsymbol{X}, \boldsymbol{z})$ to parameterize $q(\boldsymbol{z}|\boldsymbol{X}, \boldsymbol{F})$, $p(\boldsymbol{z}|\boldsymbol{X})$, and $p(\boldsymbol{F}|\boldsymbol{X}, \boldsymbol{z})$, respectively. Here, $\phi$, $\varphi$, and $\psi$ denote the model parameters of the Transformer encoder, the posterior network, and the prior network, respectively, and $\theta$ comprises the parameters of the Transformer decoder and the feature fusion module. Therefore, training the CVAE with reconstruction loss and KL divergence is equivalent to maximizing the lower bound, i.e., maximizing the conditional distribution $p(\boldsymbol{F} \mid \boldsymbol{X})$.

## C    T-SNE DATASET CONSTRUCTION AND SUPPLEMENTARY EXPERIMENTS

### C.1    T-SNE DATASET CONSTRUCTION

In Fig. 4 and Fig. 5, we constructed a dedicated dataset to analyze the separability of function categories and the continuity of numerical features in the latent space. The dataset covers three representative classes of functions: exponential functions, trigonometric functions, and logarithmic functions. The exponential class includes the operator $\exp(\cdot)$, the trigonometric class includes $\sin(\cdot)$, $\cos(\cdot)$, and $\tan(\cdot)$, while the logarithmic class uses $\log(\cdot)$.

To systematically examine the impact of input dimensionality on latent space distribution, we generated two cases for each function class: 2-dimensional input ($D = 2$) and 5-dimensional input ($D = 5$). This setting allows us to verify whether the latent space maintains clear structural separability and numerical continuity under both low- and higher-dimensional inputs.

The expression generation process is as follows: first, we specify the dimensionality of the variable x, and then combine the basic operators $+, -, \times, \div$ with the corresponding function operators from each category. To ensure comparability, we constrain the expression length such that the number of tokens (including coefficients, operators, and variables) is less than 25.

In terms of scale, for each function class and dimensionality combination, we generated 5000 equations, and for each equation, 200 data points were sampled to construct the corresponding dataset. This design provides sufficient statistical coverage and ensures that the t-SNE visualization results are representative.

In summary, this dataset systematically spans function categories, input dimensionalities, expression structures, and sample sizes, serving as a solid basis for validating the global structural separability and local numerical feature continuity of the latent space.

## C.2    INTERPOLATION RESULTS IN LATENT SPACE

Table 1: Linear interpolation between two expressions ($1\rightarrow2$) in latent space across methods. For each pair, we decode from $z(ratio) = (1-ratio)z_1 + ratio\, z_2$ with $ratio \in \{0, 0.25, 0.5, 0.75, 1\}$. Expressions marked in red are syntactically invalid.

| $ratio$ | Ours | E2ESR | SNIP |
|---|---|---|---|
| **Pair 1:** Expression 1: $c\exp(x)\cdot\sin(x-c) \rightarrow$ Expression 2: $\log(x+c)\cdot\cos(x)$ | | | |
| 0 | $c\exp(x)\sin(x-c)$ | $c\exp(x)\sin(x-c)$ | $c\exp(x)\sin(x-c)$ |
| 0.25 | $c\exp(x)\sin(x-c) + c\sin(x)$ | Invalid | $\exp(x+c)\sin(x) + c\sin(x)$ |
| 0.5 | $c\exp(c\cdot c)\sin(x) + \log(c\cdot x)\cos(c)$ | Invalid | $\exp(x)\cos(x) + c\cos(x+c)$ |
| 0.75 | $\exp(c) + \log(x+c)\cos(x)$ | $\log(x)\sin(x)\cdot\cos(x)$ | Invalid |
| 1 | $\log(x+c)\cos(x)$ | $\log(x+c)\cos(x)$ | $\log(x+c)\cos(x)$ |
| **Pair 2:** Expression 1: $x\cos(c\,x) + c \rightarrow$ Expression 2: $x - \sin(x-c)$ | | | |
| 0 | $x\cos(c\,x) + c$ | $x\cos(c\,x) + c$ | $x\cos(c\,x) + c$ |
| 0.25 | $x\cos(c\,x) + c - \cos(x)$ | Invalid | $x\cos(c) + c\sin(x)$ |
| 0.5 | $c\cos(c\cdot x) + c - \cos(x-c)$ | Invalid | Invalid |
| 0.75 | $cos(c\cdot x) - \sin(x-c)$ | Invalid | Invalid |
| 1 | $x - \sin(x-c)$ | $x - \sin(x-c)$ | $x - \sin(x-c)$ |

We compare the continuous interpolation capabilities of generative (our GenSR) and discriminative (such as SNIP and E2E) latent spaces through interpolation experiments in the latent space. For each method, we first feed the numerical samples of Expression 1 into its numeric encoding branch. In GenSR, this branch corresponds to the prior branch that only uses numeric inputs, while in E2ESR and SNIP it is implemented as their respective numeric encoders. The output of this branch is taken as the latent representation $z_1$. Expression 2 is handled in the same manner to obtain $z_2$.

We then construct intermediate latent points by linear interpolation,

$$z(ratio) = (1 - ratio)z_1 + ratio\, z_2$$

with $ratio \in \{0, 0.25, 0.5, 0.75, 1\}$. Each interpolated latent vector $z(ratio)$ is subsequently passed through the corresponding decoder to generate the intermediate expression. This unified procedure enables a fair comparison of different methods in terms of structural smoothness, and decoding robustness under latent-space interpolation, and allows us to assess whether their latent representations support continuous interpolation between equations.

Table 1 reports the linear interpolation between two expressions ($1\rightarrow2$) in the latent space. Our method consistently produces syntactically valid and meaningful expressions at each interpolation step, thanks to the design of a continuous generative latent space that preserves the decodability of equations under interpolation. In contrast, E2ESR and SNIP frequently yield syntactically invalid expressions (marked in red), indicating that their discriminative latent spaces lack sufficient syntactic

regularization and often degenerate into structures that cannot be converted into valid prefix forms. Moreover, the transitions generated by our method are smooth: expressions gradually transform from 1 to 2, demonstrating the structural continuity of the latent space. By comparison, E2ESR and SNIP exhibit drastic structural changes even under small variations in the latent variables, suggesting less stable latent representations. For clarity, all constants are uniformly denoted by $c$ to eliminate the influence of specific numeric values.

# D   Reconstruction Performance of Posterior vs. Prior Branch

Table 2: Reconstruction error on synthetic expression datasets. We report **Levenshtein (edit) distance** (mean $\pm$ std; lower is better) between an expression and its reconstruction. Two operator sets are considered: (i) **LogExp** $= \{+, -, \times, \div, \log, \exp\}$; (ii) **Trig** $=$ LogExp $\cup \{\sin, \cos, \tan\}$. Each dataset has 2k samples; the maximum expression length is shown in the name.

| Dataset (ops, length, vars) | Posterior branch | Prior branch |
|---|---|---|
| LogExp-15 (vars $\leq 3$) | 0.094 $(\pm 0.023)$ | 0.106 $(\pm 0.031)$ |
| LogExp-20 (vars $\leq 5$) | 0.095 $(\pm 0.027)$ | 0.112 $(\pm 0.029)$ |
| LogExp-25 (vars $\leq 5$) | 0.107 $(\pm 0.018)$ | 0.120 $(\pm 0.022)$ |
| Trig-15 (vars $\leq 3$) | 0.115 $(\pm 0.032)$ | 0.137 $(\pm 0.019)$ |
| Trig-20 (vars $\leq 5$) | 0.113 $(\pm 0.026)$ | 0.142 $(\pm 0.043)$ |
| Trig-25 (vars $\leq 5$) | 0.146 $(\pm 0.041)$ | 0.171 $(\pm 0.038)$ |

**Synthetic dataset construction.**     To systematically evaluate the reconstruction capability of our variational autoencoder, we construct six synthetic datasets under two operator sets: (i) **LogExp**, which includes basic arithmetic operators $\{+, -, \times, \div\}$ along with $\log$ and $\exp$, and (ii) **Trig**, which extends LogExp by adding trigonometric functions $\{\sin, \cos, \tan\}$. For each operator set, we generate three datasets with maximum expression lengths of 15, 20, and 25 tokens, constraining the number of variables to 3, 5, and 5, respectively. This range covers expression lengths slightly longer than those commonly seen in real-world benchmarks (the Feynman dataset has an average length of 13.56), allowing us to evaluate reconstruction performance under increasing structural complexity. Each dataset contains 2k randomly synthesized samples.

We quantify reconstruction fidelity using the Levenshtein (edit) distance, which counts the minimum number of insertions, deletions, and substitutions required to transform the reconstructed expression into the ground-truth one. This metric directly measures symbolic accuracy rather than numeric approximation quality, making it a natural choice for evaluating exact expression reconstruction.

**Results analysis.**     Table 2 reports the reconstruction performance using latent codes from both the posterior and prior branches. We summarize three consistent observations:

- **Posterior branch consistently outperforms the prior branch.** This is expected, as the posterior branch is trained by directly maximizing the expectation $\mathbb{E}_{q_\phi(\boldsymbol{z}|\boldsymbol{X},\boldsymbol{F})}[\log p_\theta(\boldsymbol{F} \mid \boldsymbol{X}, \boldsymbol{z})]$ using the ground-truth latent variables as supervision, resulting in lower edit distances and more accurate reconstructions.

- **Prior branch performs slightly worse but remains competitive.** Unlike the posterior branch, the prior branch is not trained to maximize the above expectation directly. Instead, it is optimized by aligning its output distribution with the posterior through the KL divergence loss, which leads to slightly higher edit distances. Nevertheless, the reconstruction quality remains reasonably good. This is crucial, since during inference we can only sample from the prior distribution. The results indicate that the KL loss effectively aligns the prior with the posterior, enabling accurate reconstruction even without access to ground-truth latent variables.

- **Expression complexity has a controlled impact on reconstruction.** As expression length increases and trigonometric operators are introduced, the reconstruction error grows gradually but remains well-behaved. The posterior branch maintains near-exact reconstruction even for length-25 expressions, validating that the learned latent space preserves sufficient structural information.

Overall, these results demonstrate that the learned latent space is both expressive and well-regularized: it enables faithful equation reconstruction from posterior samples, while the prior remains sufficiently aligned to support reliable generation at inference time.

# E    ABLATION STUDIES

To better understand the design choices in GenSR, we conduct ablation studies on both the latent space configuration and the CMA-ES search procedure. These experiments reveal how different hyperparameters affect accuracy, equation complexity, and runtime, and guide our choice of robust default settings.

## E.1    LATENT DIMENSION OF THE GENERATIVE SPACE

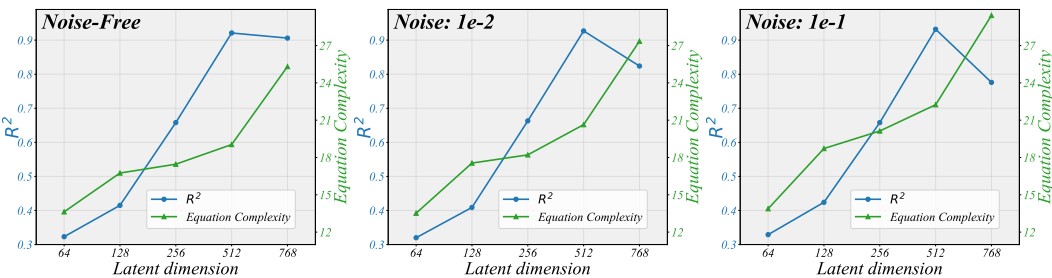

Figure 6: Effect of latent dimension in GenSR. The x-axis shows the latent dimension of the pretrained model (64, 128, 256, 512, 768), while the left and right y-axes report $R^2$ and equation complexity. Results are reported under three noise levels: Noise-Free, $1 \times 10^{-2}$, and $1 \times 10^{-1}$.

Fig. 6 presents the results of pretraining GenSR with different latent dimensions. The results in Fig. 6 To isolate the evaluation of the latent space itself, we did not apply CMA-ES optimization and instead directly compared the outputs of the pre-trained model. In the Noise-Free setting (left subplot), $R^2$ improves steadily as the latent dimension increases, reaching its maximum at 512 before showing a slight decline at 768. Equation complexity grows moderately for smaller dimensions but exhibits a marked surge beyond 512, indicating that excessively large latent spaces may encourage overly complex expressions. Under noisy conditions, the 768-dimensional model suffers from a more pronounced drop in accuracy and further inflated complexity, with performance degradation exacerbated as noise levels increase. Based on these observations, we adopt a 512-dimensional latent space for pretraining, as it offers the best trade-off and ensures robustness under noise.

## E.2    CMA-ES HYPERPARAMETERS

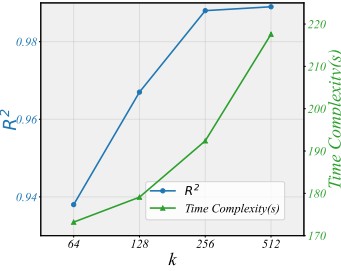

Figure 7: Effect of $k$ on CMA-ES performance.

**(1) Effect of updating principal dimensions.**    In the CMA algorithm, we adopt a diagonal approximation by retaining only the variance terms of the covariance matrix, and further update only the top-$k$ principal dimensions with the largest variances to reduce computational cost. Fig. 7 illustrates

the effect of different $k$ values. We observe that $R^2$ improves as $k$ increases and reaches a near-saturation point at $k = 256$, with almost no further gains beyond this dimension. In contrast, the search time grows approximately linearly with $k$, leading to a significant increase in computational cost for larger $k$. Balancing accuracy and efficiency, we therefore fix $k = 256$ in all subsequent experiments, as it provides the best trade-off.

**(2) Other hyperparameters.**  We further study three intrinsic hyperparameters of CMA-ES: population size $s$, initial step size $t$, and the fitness weight $\omega$ in the objective Fitness $= R^2 - \omega \cdot$complexity.

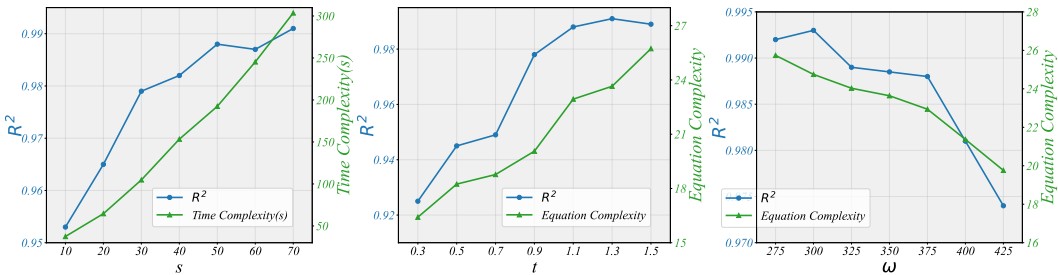

Figure 8: Ablation results on CMA-ES hyperparameters. Left: effect of population size $s$ on $R^2$ and runtime. Middle: effect of initial step size $t$ on $R^2$ and equation complexity. Right: effect of weight $\omega$ in the fitness function.

**Population size $s$.**  As $s$ increases, $R^2$ steadily improves from 0.95 to nearly 0.99 but saturates after $s = 50$, indicating that larger populations enhance global exploration yet exhibit diminishing returns. Meanwhile, runtime (i.e., time complexity) grows almost linearly from tens of seconds to nearly 300 seconds, showing a significant computational cost. Thus, medium-sized populations strike a better trade-off between accuracy and efficiency.

**Initial step size $t$.**  The value of $R^2$ rises and then falls with increasing $t$, achieving the best performance around $t = 1.1$, while equation complexity grows monotonically. Small step sizes constrain the search within local regions, whereas excessively large step sizes cause unstable convergence. A moderate $t$ therefore provides a preferable balance between accuracy and complexity.

**Fitness weight $\omega$.**  In the fitness function Fitness $= R^2 - \omega \cdot$ complexity, $\omega$ governs the trade-off between accuracy and complexity. Smaller $\omega$ emphasizes maximizing $R^2$ but yields overly complex equations, while larger $\omega$ favors simpler forms at the cost of accuracy. A moderate choice of $\omega$ effectively balances these two aspects, avoiding both overfitting and underfitting.

**Summary.**  Overall, our ablations show that optimal performance is obtained with a 512-dimensional latent space, $k = 256$ for covariance updates, and moderate settings for $s$, $t$, and $\omega$. These choices consistently yield robust trade-offs between accuracy, complexity, and runtime.

# F  MODEL ARCHITECTURE AND TRAINING CONFIGURATION

## F.1  MODEL ARCHITECTURE

As illustrated in Fig. 1, GenSR consists of two input embedders (for numerical values and equations), a shared Transformer encoder, posterior and prior MLPs, a feature fusion linear layer, and a Transformer decoder.

The numerical input $\{(\boldsymbol{x}, y)_i\}_{i=1}^m \in \mathbb{R}^{m \times (D+1)}$ is first tokenized into $\{(\boldsymbol{x}, \boldsymbol{y})_i\}_{i=1}^m \in \mathbb{R}^{m \times 3(D+1)}$, then mapped by the numerical embedder to $\boldsymbol{X}_e \in \mathbb{R}^{m \times 3(D+1) \times d_n}$, where $d_n = 64$ denotes the numerical embedding dimension. A reshape and linear projection yield the final $\boldsymbol{X} \in \mathbb{R}^{m \times d}$. Similarly, equation expressions are padded to the maximum sequence length $m$ and embedded into $\boldsymbol{F} \in \mathbb{R}^{m \times d}$ with $d = 512$.

For the posterior branch, $\boldsymbol{X}$ and $\boldsymbol{F}$ are concatenated and passed through an 8-layer Transformer encoder (8 heads, 512 hidden units), followed by an MLP to produce the posterior distribution

$\mathcal{N}(\boldsymbol{\mu}_1, \boldsymbol{\sigma}_1^2)$. For the prior branch, $\boldsymbol{X}$ alone (with masking applied) is fed into the same encoder to produce the prior distribution $\mathcal{N}(\boldsymbol{\mu}_2, \boldsymbol{\sigma}_2^2)$. Notably, the numerical input $\boldsymbol{X}$ does not include positional encodings, ensuring that only the functional relations, rather than ordering artifacts, are modeled. The KL divergence between posterior and prior distributions serves as a regularization term:

$$\mathcal{L}_{\text{KL}} = D_{\text{KL}}\big(\mathcal{N}(\boldsymbol{\mu}_1, \boldsymbol{\sigma}_1^2) \,\|\, \mathcal{N}(\boldsymbol{\mu}_2, \boldsymbol{\sigma}_2^2)\big).$$

During training, $m$ samples are drawn from the posterior $\mathcal{N}(\boldsymbol{\mu}_1, \boldsymbol{\sigma}_1^2)$ to form $\boldsymbol{Z}_1 \in \mathbb{R}^{m \times d}$, which is processed by the feature fusion linear layer and decoded by a symmetric 8-layer Transformer decoder (8 heads, 512 hidden units). The decoder generates logits of symbolic expressions, which are compared with the ground-truth equations through a reconstruction loss:

$$\mathcal{L}_{\text{CE}} = -\sum_{i=1}^{m} \log p_\theta(f_i \mid \boldsymbol{z}_1^i),$$

where $f_i$ denotes the $i$-th token of the target equation and $\theta$ the decoder parameters.

The overall training objective combines these two terms:

$$\mathcal{L} = \mathcal{L}_{\text{CE}} + \lambda\, \mathcal{L}_{\text{KL}},$$

where $\lambda$ is a balancing hyperparameter. We employ KL annealing to gradually increase $\lambda$ from 0 to its full value during training. This prevents the KL term from dominating the optimization in the early stages and mitigates posterior collapse, while ensuring a well-structured latent space in later stages.

During the inference phase, we use the prior distribution $\mathcal{N}(\boldsymbol{\mu}_2, \boldsymbol{\sigma}_2^2)$ to form latent variables $\boldsymbol{Z}_2 \in \mathbb{R}^{m \times d}$, which are then fed into the Transformer decoder to generate candidate equations. Each equation is evaluated with a fitness score, which is subsequently utilized by the CMA-ES optimization procedure to iteratively refine the search and obtain the final predicted equation.

### F.2 Training Configuration

We adopt the Adam optimizer with a batch size of 256 and train the model for 200 epochs, each consisting of 1000 steps. The learning rate is set to $1.0 \times 10^{-3}$ and follows the Noam schedule with $w = 8000$ warm-up steps. This schedule allows the learning rate to increase linearly during warm-up and then decay proportionally to the inverse square root of the step number. For the KL divergence weight $\lambda$, we apply an annealing strategy: $\lambda$ is linearly increased from 0 to 1.0 during the first 50% of training steps and kept constant thereafter. This prevents the KL term from dominating optimization at the beginning and mitigates posterior collapse, while ensuring a structured latent space in later stages. The model comprises 162.1 million trainable parameters. All experiments are conducted using two NVIDIA H800 GPUs (Hopper architecture, 80GB memory each), with each epoch requiring approximately one hour of training time.

## G  Detailed Experiment Results

### G.1  Detailed Description of Evaluation Datasets

For our experimental evaluation, we adopt the SRBench benchmark (La Cava et al., 2021), a widely used and challenging collection of datasets for symbolic regression. SRBench consists of three representative groups: the Feynman dataset, the Strogatz dataset, and the Black-box dataset. These datasets differ in input dimensionality, sample size, functional properties, and task difficulty, jointly forming a comprehensive evaluation framework for symbolic regression methods and providing a standardized platform for comparison.

**Feynman dataset.** The Feynman dataset (Udrescu et al., 2020) is derived from the Feynman Lectures on Physics and contains 119 equations spanning diverse domains such as mechanics, electromagnetism, and quantum physics. These equations are structurally diverse and grounded in real physical laws, making them a key benchmark for evaluating symbolic regression in scientific discovery. To ensure consistency across tasks, the dataset constrains the input dimensionality to $D \leq 10$.

A unique advantage is that the true underlying functions are fully available, eliminating the ambiguity present in black-box tasks. With a cumulative size of approximately $10^5$ data points, this dataset allows us to assess both large-scale regression accuracy and generalization across diverse physical laws.

**Strogatz dataset.** The Strogatz dataset (Strogatz, 2018) originates from nonlinear dynamical systems, inspired by the classical ODE-Strogatz collection. Each task corresponds to a two-dimensional dynamical system modeling problem, with input dimensionality fixed at $D = 2$. Unlike Feynman, which emphasizes physical interpretability, the Strogatz dataset focuses on dynamic behaviors and is particularly suited to testing the ability of methods to handle nonlinearity and temporal dependencies. Each sub-dataset consists of approximately $N = 400$ samples, with true underlying functions provided to enable evaluation within a well-defined dynamical framework. This dataset is thus critical for benchmarking performance under small-sample and high-nonlinearity conditions.

**Black-box dataset.** The Black-box dataset (Olson et al., 2017) highlights the applicability of symbolic regression to complex real-world scenarios. This group is primarily sourced from the PMLB repository and has been widely adopted in SRBench. Unlike Feynman and Strogatz, the target functions in this dataset are unknown, mimicking real-world regression tasks. These datasets often contain higher levels of noise and uncertainty, making them an effective test of robustness. To ensure comparability, input dimensionality is limited to $D \leq 10$. The collection comprises 57 sub-datasets, covering both real-world and synthetic cases, with dataset sizes ranging from just a few dozen to tens of thousands of samples. This wide variation tests not only the accuracy of algorithms but also their stability and generalization under noisy conditions.

In summary, the Feynman dataset emphasizes physical interpretability, the Strogatz dataset captures the complexity of nonlinear dynamical systems, and the Black-box dataset stresses robustness under real-world uncertainty and noise. Together, these three datasets form a complementary and multi-faceted evaluation environment for symbolic regression. Under this benchmark, we are able to comprehensively examine trade-offs in accuracy, efficiency, and complexity, while highlighting the robustness and generality of our proposed approach across diverse tasks.

## G.2 EVALUATION METRICS

To evaluate the performance of symbolic regression methods, we consider three key metrics that capture accuracy, efficiency, and interpretability:

- **Coefficient of determination** $(R^2)$**:** This metric quantifies how well the discovered symbolic expression explains the variance of the target data. It is formally defined as:

$$R^2 = 1 - \frac{\sum_{i=1}^{n}(y_i - \hat{y}_i)^2}{\sum_{i=1}^{n}(y_i - \bar{y})^2}, \tag{2}$$

  where $y_i$ denotes the ground-truth values, $\hat{y}_i$ the predicted values, and $\bar{y}$ the mean of the ground-truth values. A higher $R^2$ indicates better predictive accuracy, with $R^2 = 1$ representing a perfect fit and $R^2 = 0$ implying no improvement over the mean predictor.

- **Time complexity:** This metric measures the wall-clock time required for the algorithm to complete the symbolic regression task and output a final expression, reported in seconds (s). It reflects the computational efficiency of the method, where shorter search times indicate faster convergence and reduced resource consumption.

- **Equation complexity:** This metric evaluates the interpretability and compactness of the discovered expressions. We define complexity as the total number of tokens in the final equation:

$$\text{Complexity}(f) = \sum_{j=1}^{m} \mathbb{I}(t_j), \tag{3}$$

  where $f$ denotes the discovered equation, $\{t_j\}_{j=1}^{m}$ are its tokens (including coefficients, operators, and variables), and $\mathbb{I}(\cdot)$ is an indicator function that counts the presence of each token. Lower complexity corresponds to simpler and more human-readable expressions, which are less prone to overfitting and closer to the underlying physical or mathematical laws.

### G.3 BASELINES

**Evolutionary algorithm-based symbolic regression.** We include several genetic programming variants: GPlearn (Stephens et al., 2015), a scikit-learn–style GP framework; AFP (Schmidt & Lipson, 2010), which maintains diversity via age-fitness Pareto optimization; ITEA (de Franca & Aldeia, 2021), which exploits interaction–transformation operators; EPLEX (La Cava et al., 2016), based on epsilon-lexicase selection; MRGP (Arnaldo et al., 2014), combining multiple regression with GP; SBP-GP (Virgolin et al., 2019), using semantic backpropagation with linear scaling; GP-GOMEA (Virgolin et al., 2021), a model-based GP with efficient linkage learning; Operon (Burlacu et al., 2020), a high-performance C++ SR framework; and FEAT (La Cava et al., 2019), which evolves networks of trees to learn concise regression models.

**Deep/Neural and reinforcement learning-based symbolic regression.** We further consider neural and RL-driven methods: SPL (Sun et al.), which employs Monte Carlo tree search for physics discovery; DSR (Petersen et al.), a reinforcement learning method with risk-seeking policy gradients; RSRM (Xu et al., 2024), a reinforcement symbolic regression machine; TPSR (Shojaee et al., 2023), which formulates symbolic regression as a Transformer-based planning problem; E2ESR (Kamienny et al., 2022), an end-to-end Transformer framework; SNIP (Meidani et al.), which unifies symbolic and numeric pretraining; RAG-SR (Zhang et al., 2025a), leveraging retrieval-augmented generation; MDLformer (Yu et al., 2025), guided by the minimum description length principle; and NeurSR (Biggio et al., 2021), a scalable neural symbolic regression approach.

**Physics- and probabilistic-inspired methods.** We also evaluate AI Feynman (Udrescu et al., 2020), which exploits physics priors and graph modularity for symbolic discovery, and BSR (Jin et al., 2020), a Bayesian symbolic regression approach that provides uncertainty estimation.

**Conventional machine learning baselines.** Finally, we compare against strong non-symbolic baselines, including tree-based ensemble methods: XGBoost (Chen & Guestrin, 2016), LightGBM (Ke et al., 2017), AdaBoost (Freund & Schapire, 1997), and Random Forest (Rigatti, 2017).

### G.4 FUNCTION FAMILY CLASSIFICATION TASK

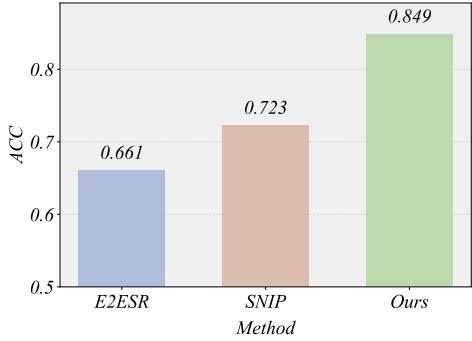

Figure 9: Accuracy comparison of different methods on the classification task. GenSR (Ours) achieves the highest accuracy (0.849), significantly outperforming E2ESR (0.661) and SNIP (0.723).

To further examine the transferability of the learned generative latent space beyond symbolic regression, we design a function family classification task. The goal of this experiment is to assess whether the latent space representations learned by GenSR capture not only structural continuity and numerical smoothness, but also discriminative features that support downstream classification.

**Dataset construction.** We synthesize a dataset covering five representative function families: exponential, logarithmic, sine, cosine, and tangent. To increase diversity and difficulty, each family is instantiated under four different input dimensionalities (2D, 3D, 4D, and 5D), leading to a total of $5 \times 4 = 20$ classes. For each class, we randomly generate 2000 samples, where each sample consists

of input points and their corresponding function values. The dataset is split into training, validation, and test sets with a ratio of 3:1:1 to ensure fair and stable evaluation.

**Model and training setup.** We adopt the GenSR encoder pretrained on symbolic regression, which produces 512-dimensional latent vectors. On top of the encoder, we attach a lightweight classification head composed of two fully connected layers: the first layer maps the latent vector to a hidden representation with ReLU activation, and the second layer outputs a 20-dimensional logits vector followed by a softmax layer to obtain class probabilities. We fine-tune the entire network using cross-entropy loss with the Adam optimizer, a learning rate of $1 \times 10^{-3}$, and a batch size of 128. Training is run for at most 100 epochs, with early stopping if no improvement is observed on the validation set for five consecutive epochs. At test time, the class with the highest probability is selected as the prediction, and overall accuracy is reported.

**Comparison and results.** For a fair comparison, we apply the same classification head design, data split, and training procedure to the encoders of E2ESR (Kamienny et al., 2022) and SNIP (Meidani et al.). As shown in Fig. 9, GenSR achieves significantly higher classification accuracy than both baselines. This result demonstrates that the latent space learned by GenSR is not only generative and structurally continuous, but also discriminative and generalizable, providing a strong foundation for downstream tasks beyond symbolic regression. In other words, GenSR yields representations that are simultaneously useful for generation, search, and classification, highlighting its potential for broader applications.

In the future, the latent space obtained by the CVAE also can be applied to a variety of domains, such as biological contexts (Duan et al., 2025), psychological studies (Jia et al., 2025), energy systems and meteorological forecasting where spatiotemporal patterns play a critical role (Hu et al., 2024; 2025), and broader areas of artificial intelligence where interpretability and robustness are of particular importance (Li et al., 2024; 2023b;a; Huang et al., 2026; Feng et al., 2026; 2025; Kang et al., 2025; Zhang et al., 2025b; Yang et al.).

### G.5    COMPLETE NUMERICAL RESULTS

Tables 4, 5, 6, and 7 report the performance of different methods on the Strogatz (Strogatz, 2018) and Feynman datasets (Udrescu et al., 2020) under varying noise levels, while Table 3 summarizes the results on the Black-box datasets (Olson et al., 2017). Across all these benchmarks, our method demonstrates a consistently strong balance among the three key metrics—accuracy ($R^2$), search efficiency (time), and equation complexity. Notably, even under noisy conditions, our approach maintains high $R^2$ scores with smaller variance, achieves shorter runtime, and yields simpler equations, highlighting its robustness and overall superiority compared to existing baselines.

Table 3: Complete experimental results of black-box dataset.

| Method | Test $R^2$ ↑ | Complexity ↓ | Time (s) ↓ |
|---|---|---|---|
| Operon | 0.7945 | 65.69 | 2974 |
| GP-GOMEA | 0.7381 | 30.27 | 9636 |
| MDL | 0.6258 | 29.88 | 541.7 |
| DSR | 0.5625 | 9.465 | 36852 |
| SBP-GP | 0.7869 | 634 | 149344 |
| FEAT | 0.7621 | 82.49 | 6432 |
| EPLEX | 0.7372 | 53.14 | 15796 |
| AFP-FE | 0.6400 | 36.04 | 6184 |
| AFP | 0.6333 | 34.89 | 6033 |
| GPlearn | 0.5390 | 19.06 | 24254 |
| Linear | 0.4437 | 17.4 | 0.2447 |
| NeurSR | 0.1228 | 13.33 | 11.7 |
| XGB | 0.7496 | 20186 | 236 |
| AdaBoost | 0.6939 | 9481 | 65.12 |
| LGBM | 0.6410 | 5734 | 29.9 |
| ITEA | 0.6295 | 116.7 | 12183 |
| SNIP | 0.3335 | 38.91 | 3.286 |
| BSR | 0.2725 | 22.52 | 59822 |
| RandomForest | 0.6615 | 1517178 | 120.4 |
| KernelRidge | 0.5952 | 1824 | 39.19 |
| FFX | 0.5575 | 1562 | 244.3 |
| E2ESR | 0.3612 | 61.09 | 7.101 |
| MRGP | 0.5300 | 10802 | 165007 |
| MLP | 0.5238 | 3882 | 30.49 |
| AIFeynman2 | 0.2110 | 2240 | 86854 |
| SPL | 0.5472 | 12.96 | 2047 |
| RSRM | 0.3324 | 9.23 | 1752 |
| **Ours** | 0.8416 | 35.05 | 2725 |

Table 4: Complete experimental results at a noise level $\epsilon = 0.0$. The results include test set $R^2$, search time, and formula complexity of different methods in the Feynman and Strogatz datasets.

| Method | Strogatz Dataset ($\epsilon = 0.0$) | | | Feynman Dataset ($\epsilon = 0.0$) | | |
|---|---|---|---|---|---|---|
| | $R^2\uparrow$ | Time (s)$\downarrow$ | Complx$\downarrow$ | $R^2\uparrow$ | Time (s)$\downarrow$ | Complx$\downarrow$ |
| **FEAT** | $0.9210_{(\pm0.054)}$ | $1135_{(\pm970)}$ | $119_{(\pm16)}$ | $0.9190_{(\pm0.014)}$ | $2417_{(\pm1737)}$ | $205.3_{(\pm13)}$ |
| **NeurSR** | $0.5206_{(\pm0.029)}$ | $14.82_{(\pm1.7)}$ | $11.3_{(\pm0.51)}$ | $0.3958_{(\pm0.013)}$ | $24.18_{(\pm2.2)}$ | $13.28_{(\pm0.15)}$ |
| **E2ESR** | $0.5341_{(\pm0.043)}$ | $3.729_{(\pm2)}$ | $32.49_{(\pm2.9)}$ | $0.8570_{(\pm0.013)}$ | $4.062_{(\pm1.3)}$ | $36.02_{(\pm0.88)}$ |
| **SNIP** | $0.9952_{(\pm0.002)}$ | $509_{(\pm21)}$ | $29_{(\pm1.2)}$ | $0.985_{(\pm0.004)}$ | $1203.5_{(\pm145)}$ | $31.71_{(\pm1.8)}$ |
| **GPlearn** | $0.7689_{(\pm0.067)}$ | $1195_{(\pm592)}$ | $28.96_{(\pm2.8)}$ | $0.8809_{(\pm0.027)}$ | $3900_{(\pm1052)}$ | $72.43_{(\pm29)}$ |
| **AFP** | $0.9248_{(\pm0.041)}$ | $192.2_{(\pm106)}$ | $38.17_{(\pm2.8)}$ | $0.9590_{(\pm0.005)}$ | $3655_{(\pm1999)}$ | $36.87_{(\pm2.2)}$ |
| **AFP-FE** | $0.9442_{(\pm0.045)}$ | $11041_{(\pm14277)}$ | $46.16_{(\pm4.1)}$ | $0.9806_{(\pm0.007)}$ | $17817_{(\pm530)}$ | $39.97_{(\pm1.9)}$ |
| **EPLEX** | $0.8125_{(\pm0.065)}$ | $548.2_{(\pm258)}$ | $50.09_{(\pm2.7)}$ | $0.9869_{(\pm0.006)}$ | $12771_{(\pm4998)}$ | $52.95_{(\pm1.3)}$ |
| **SBP-GP** | $0.9812_{(\pm0.016)}$ | $17591_{(\pm1765)}$ | $712.2_{(\pm39)}$ | $0.9945_{(\pm0.001)}$ | $28901_{(\pm37)}$ | $489.4_{(\pm16)}$ |
| **GP-GOMEA** | $0.9925_{(\pm0.009)}$ | $2760_{(\pm1258)}$ | $36.43_{(\pm2.4)}$ | $0.9956_{(\pm0.003)}$ | $5030_{(\pm967)}$ | $34.57_{(\pm1.5)}$ |
| **Operon** | $0.9878_{(\pm0.023)}$ | $66.58_{(\pm10)}$ | $59.23_{(\pm2.1)}$ | $0.9889_{(\pm0.006)}$ | $2174_{(\pm373)}$ | $69.88_{(\pm1.8)}$ |
| **SPL** | $0.7390_{(\pm0.047)}$ | $322.1_{(\pm180)}$ | $14.55_{(\pm2.5)}$ | $0.7073_{(\pm0.011)}$ | $209_{(\pm145)}$ | $12.88_{(\pm0.57)}$ |
| **DSR** | $0.7602_{(\pm0.086)}$ | $1858_{(\pm2617)}$ | $15.6_{(\pm1.7)}$ | $0.8441_{(\pm0.091)}$ | $1733_{(\pm3105)}$ | $14.86_{(\pm1)}$ |
| **RSRM** | $0.5501_{(\pm0.103)}$ | $121.9_{(\pm36)}$ | $13.09_{(\pm2.3)}$ | $0.8003_{(\pm0.013)}$ | $116.3_{(\pm31)}$ | $13.17_{(\pm0.47)}$ |
| **AIFeynman2** | $0.6459_{(\pm0.039)}$ | $762.1_{(\pm424)}$ | $22.26_{(\pm1.7)}$ | $0.9314_{(\pm0.016)}$ | $854.3_{(\pm24)}$ | $124.5_{(\pm16)}$ |
| **BSR** | $0.8455_{(\pm0.044)}$ | $31380_{(\pm23952)}$ | $38.98_{(\pm5.4)}$ | $0.6609_{(\pm0.018)}$ | $29065_{(\pm765)}$ | $25.5_{(\pm0.23)}$ |
| **MDL** | $0.9900_{(\pm0.009)}$ | $186.6_{(\pm35)}$ | $14.07_{(\pm1.9)}$ | $0.9171_{(\pm0.005)}$ | $467.3_{(\pm415)}$ | $23.4_{(\pm1.2)}$ |
| **TPSR** | $0.965_{(\pm0.024)}$ | $291_{(\pm53)}$ | $56.2_{(\pm1.4)}$ | $0.9921_{(\pm0.002)}$ | $236_{(\pm39)}$ | $57.27_{(\pm1.9)}$ |
| **RAG-SR** | $0.9914_{(\pm0.006)}$ | $1477_{(\pm305)}$ | $46.5_{(\pm3.1)}$ | $0.9926_{(\pm0.002)}$ | $3156_{(\pm305)}$ | $46.51_{(\pm1.4)}$ |
| **Ours** | $0.9918_{(\pm0.003)}$ | $33_{(\pm5.7)}$ | $20.53_{(\pm0.9)}$ | $0.9872_{(\pm0.003)}$ | $192.3_{(\pm21)}$ | $22.94_{(\pm0.5)}$ |

Table 5: Complete experimental results at a noise level $\epsilon = 0.001$. The results include test set $R^2$, search time, and formula complexity of different methods in the Feynman and Strogatz datasets.

| Method | Strogatz Dataset ($\epsilon = 0.001$) | | | Feynman Dataset ($\epsilon = 0.001$) | | |
|---|---|---|---|---|---|---|
| | $R^2\uparrow$ | Time (s)$\downarrow$ | Complx$\downarrow$ | $R^2\uparrow$ | Time (s)$\downarrow$ | Complx$\downarrow$ |
| **FEAT** | $0.9244_{(\pm0.032)}$ | $594.4_{(\pm181)}$ | $106.7_{(\pm15)}$ | $0.9207_{(\pm0.006)}$ | $1726_{(\pm242)}$ | $196.5_{(\pm12)}$ |
| **NeurSR** | $0.5219_{(\pm0.031)}$ | $15.07_{(\pm1.7)}$ | $11.41_{(\pm0.31)}$ | $0.3979_{(\pm0.013)}$ | $24.51_{(\pm1.7)}$ | $13.24_{(\pm0.13)}$ |
| **E2ESR** | $0.5105_{(\pm0.060)}$ | $3.436_{(\pm0.75)}$ | $33.83_{(\pm4.4)}$ | $0.8585_{(\pm0.010)}$ | $3.894_{(\pm0.98)}$ | $35.85_{(\pm1.5)}$ |
| **SNIP** | $0.9667_{(\pm0.024)}$ | $277.9_{(\pm18.4)}$ | $27.57_{(\pm1.3)}$ | $0.9904_{(\pm0.002)}$ | $1365.07_{(\pm104)}$ | $31.43_{(\pm0.23)}$ |
| **GPlearn** | $0.7955_{(\pm0.067)}$ | $913.3_{(\pm121)}$ | $29.59_{(\pm2.5)}$ | $0.8902_{(\pm0.008)}$ | $3316_{(\pm540)}$ | $60.49_{(\pm12)}$ |
| **AFP** | $0.9172_{(\pm0.052)}$ | $143.4_{(\pm27)}$ | $38.75_{(\pm4.6)}$ | $0.9606_{(\pm0.006)}$ | $3711_{(\pm457)}$ | $39.33_{(\pm1.6)}$ |
| **AFP-FE** | $0.9447_{(\pm0.042)}$ | $8108_{(\pm584)}$ | $48.74_{(\pm3)}$ | $0.9805_{(\pm0.007)}$ | $26160_{(\pm157)}$ | $46.47_{(\pm1.2)}$ |
| **EPLEX** | $0.8488_{(\pm0.053)}$ | $416.1_{(\pm81)}$ | $49.26_{(\pm4.7)}$ | $0.9866_{(\pm0.007)}$ | $12341_{(\pm436)}$ | $56.03_{(\pm1.3)}$ |
| **SBP-GP** | $0.9879_{(\pm0.015)}$ | $19596_{(\pm1233)}$ | $820.5_{(\pm41)}$ | $0.9953_{(\pm0.001)}$ | $28940_{(\pm20)}$ | $574.4_{(\pm13)}$ |
| **GP-GOMEA** | $0.9914_{(\pm0.009)}$ | $804_{(\pm603)}$ | $41.14_{(\pm2.9)}$ | $0.9962_{(\pm0.001)}$ | $2904_{(\pm146)}$ | $45.23_{(\pm0.71)}$ |
| **Operon** | $0.9843_{(\pm0.036)}$ | $75.68_{(\pm14)}$ | $67.03_{(\pm5.5)}$ | $0.9916_{(\pm0.005)}$ | $2195_{(\pm404)}$ | $69.67_{(\pm1.6)}$ |
| **SPL** | $0.7526_{(\pm0.049)}$ | $358.8_{(\pm211)}$ | $14.4_{(\pm2.5)}$ | $0.7073_{(\pm0.016)}$ | $275.5_{(\pm206)}$ | $13.15_{(\pm0.44)}$ |
| **DSR** | $0.8067_{(\pm0.048)}$ | $500.8_{(\pm317)}$ | $18.66_{(\pm2.2)}$ | $0.8764_{(\pm0.003)}$ | $830.1_{(\pm282)}$ | $16.04_{(\pm0.31)}$ |
| **RSRM** | $0.5447_{(\pm0.105)}$ | $129.7_{(\pm8.1)}$ | $12.23_{(\pm0.65)}$ | $0.8104_{(\pm0.025)}$ | $128.2_{(\pm3.8)}$ | $13.04_{(\pm0.43)}$ |
| **AIFeynman2** | $0.6855_{(\pm0.091)}$ | $84.19_{(\pm74)}$ | $25.64_{(\pm3)}$ | $0.9177_{(\pm0.008)}$ | $638_{(\pm21)}$ | $130.6_{(\pm17)}$ |
| **BSR** | $0.8224_{(\pm0.121)}$ | $24299_{(\pm3478)}$ | $37.68_{(\pm2.2)}$ | $0.6538_{(\pm0.023)}$ | $30255_{(\pm4770)}$ | $25.85_{(\pm0.57)}$ |
| **MDL** | $0.9965_{(\pm0.004)}$ | $171.5_{(\pm30)}$ | $21.38_{(\pm1.4)}$ | $0.9079_{(\pm0.008)}$ | $428.9_{(\pm261)}$ | $32.35_{(\pm1.1)}$ |
| **TPSR** | $0.9216_{(\pm0.052)}$ | $193.22_{(\pm24.8)}$ | $55.64_{(\pm3.1)}$ | $0.992_{(\pm0.001)}$ | $190.2_{(\pm7.9)}$ | $59.75_{(\pm2.6)}$ |
| **RAG-SR** | $0.9908_{(\pm0.008)}$ | $591.4_{(\pm38.4)}$ | $49.29_{(\pm2.1)}$ | $0.9917_{(\pm0.003)}$ | $3590_{(\pm235)}$ | $49.42_{(\pm1.1)}$ |
| **Ours** | $0.9951_{(\pm0.007)}$ | $41.56_{(\pm6.1)}$ | $20.45_{(\pm0.8)}$ | $0.9883_{(\pm0.001)}$ | $279.31_{(\pm32.6)}$ | $23.5_{(\pm0.7)}$ |

Table 6: Complete experimental results at a noise level $\epsilon = 0.01$. The results include test set $R^2$, search time, and formula complexity of different methods in the Feynman and Strogatz datasets.

| Method | Strogatz Dataset ($\epsilon = 0.01$) | | | Feynman Dataset ($\epsilon = 0.01$) | | |
|---|---|---|---|---|---|---|
| | $R^2\uparrow$ | Time (s)$\downarrow$ | Complx$\downarrow$ | $R^2\uparrow$ | Time (s)$\downarrow$ | Complx$\downarrow$ |
| FEAT | $0.9244_{(\pm 0.043)}$ | $472.9_{(\pm 90)}$ | $95.61_{(\pm 16)}$ | $0.9212_{(\pm 0.010)}$ | $1464_{(\pm 365)}$ | $167.1_{(\pm 6.5)}$ |
| NeurSR | $0.5179_{(\pm 0.042)}$ | $15.48_{(\pm 1.4)}$ | $11.63_{(\pm 0.34)}$ | $0.3942_{(\pm 0.011)}$ | $24.62_{(\pm 1.7)}$ | $13.26_{(\pm 0.12)}$ |
| E2ESR | $0.5031_{(\pm 0.034)}$ | $3.392_{(\pm 0.72)}$ | $35.94_{(\pm 1.8)}$ | $0.8345_{(\pm 0.007)}$ | $4.274_{(\pm 0.58)}$ | $40.07_{(\pm 0.90)}$ |
| SNIP | $0.9844_{(\pm 0.025)}$ | $237.61_{(\pm 32)}$ | $29.35_{(\pm 1.8)}$ | $0.987_{(\pm 0.007)}$ | $1210.1_{(\pm 154)}$ | $32.62_{(\pm 1.3)}$ |
| GPlearn | $0.7956_{(\pm 0.059)}$ | $907.4_{(\pm 110)}$ | $30.59_{(\pm 4.3)}$ | $0.8890_{(\pm 0.009)}$ | $3351_{(\pm 437)}$ | $60.07_{(\pm 19)}$ |
| AFP | $0.9153_{(\pm 0.053)}$ | $152.2_{(\pm 15)}$ | $38.62_{(\pm 7.1)}$ | $0.9610_{(\pm 0.005)}$ | $4090_{(\pm 758)}$ | $40.86_{(\pm 1.1)}$ |
| AFP-FE | $0.9582_{(\pm 0.041)}$ | $8898_{(\pm 579)}$ | $48.8_{(\pm 5.4)}$ | $0.9819_{(\pm 0.008)}$ | $27763_{(\pm 297)}$ | $46.92_{(\pm 2.3)}$ |
| EPLEX | $0.8562_{(\pm 0.040)}$ | $437.6_{(\pm 64)}$ | $53.07_{(\pm 3.8)}$ | $0.9910_{(\pm 0.002)}$ | $11043_{(\pm 718)}$ | $54_{(\pm 0.68)}$ |
| SBP-GP | $0.9813_{(\pm 0.015)}$ | $20783_{(\pm 776)}$ | $850.9_{(\pm 34)}$ | $0.9950_{(\pm 0.001)}$ | $28954_{(\pm 15)}$ | $595.5_{(\pm 12)}$ |
| GP-GOMEA | $0.9783_{(\pm 0.029)}$ | $765.6_{(\pm 1005)}$ | $42.64_{(\pm 4.5)}$ | $0.9967_{(\pm 0.001)}$ | $3020_{(\pm 360)}$ | $44.67_{(\pm 1.1)}$ |
| Operon | $0.9829_{(\pm 0.031)}$ | $94.92_{(\pm 15)}$ | $81.68_{(\pm 1.2)}$ | $0.9878_{(\pm 0.010)}$ | $3165_{(\pm 549)}$ | $87.96_{(\pm 1.3)}$ |
| SPL | $0.7388_{(\pm 0.060)}$ | $413.3_{(\pm 295)}$ | $14.71_{(\pm 1.8)}$ | $0.7133_{(\pm 0.007)}$ | $295.1_{(\pm 175)}$ | $13.43_{(\pm 0.58)}$ |
| DSR | $0.8199_{(\pm 0.055)}$ | $492.9_{(\pm 287)}$ | $18.51_{(\pm 1.2)}$ | $0.8782_{(\pm 0.004)}$ | $929.8_{(\pm 422)}$ | $16.2_{(\pm 0.41)}$ |
| RSRM | $0.5969_{(\pm 0.077)}$ | $142.2_{(\pm 21)}$ | $14.22_{(\pm 1.5)}$ | $0.8092_{(\pm 0.015)}$ | $131.6_{(\pm 14)}$ | $12.97_{(\pm 0.34)}$ |
| AIFeynman2 | $0.7753_{(\pm 0.047)}$ | $85.17_{(\pm 75)}$ | $32.41_{(\pm 4.1)}$ | $0.8732_{(\pm 0.021)}$ | $629.4_{(\pm 5.9)}$ | $155.2_{(\pm 8)}$ |
| BSR | $0.8127_{(\pm 0.070)}$ | $23622_{(\pm 554)}$ | $38.74_{(\pm 2.6)}$ | $0.6734_{(\pm 0.018)}$ | $30411_{(\pm 4711)}$ | $28.03_{(\pm 0.49)}$ |
| MDL | $0.9718_{(\pm 0.057)}$ | $505.1_{(\pm 34)}$ | $20.31_{(\pm 0.81)}$ | $0.9140_{(\pm 0.009)}$ | $844.3_{(\pm 561)}$ | $31.15_{(\pm 1.5)}$ |
| TPSR | $0.9798_{(\pm 0.028)}$ | $175.09_{(\pm 19)}$ | $56.14_{(\pm 3.5)}$ | $0.9911_{(\pm 0.004)}$ | $217.06_{(\pm 34)}$ | $64.27_{(\pm 2.6)}$ |
| RAG-SR | $0.9867_{(\pm 0.019)}$ | $509_{(\pm 40)}$ | $49.43_{(\pm 1.7)}$ | $0.9905_{(\pm 0.007)}$ | $3241_{(\pm 417)}$ | $72.41_{(\pm 2.2)}$ |
| Ours | $0.9936_{(\pm 0.007)}$ | $44.58_{(\pm 24)}$ | $20.42_{(\pm 0.6)}$ | $0.9872_{(\pm 0.002)}$ | $486.83_{(\pm 48)}$ | $23.36_{(\pm 0.7)}$ |

Table 7: Complete experimental results at a noise level $\epsilon = 0.1$. The results include test set $R^2$, search time, and formula complexity of different methods in the Feynman and Strogatz datasets.

| Method | Strogatz Dataset ($\epsilon = 0.1$) | | | Feynman Dataset ($\epsilon = 0.1$) | | |
|---|---|---|---|---|---|---|
| | $R^2\uparrow$ | Time (s)$\downarrow$ | Complx$\downarrow$ | $R^2\uparrow$ | Time (s)$\downarrow$ | Complx$\downarrow$ |
| FEAT | $0.9228_{(\pm 0.027)}$ | $446.9_{(\pm 73)}$ | $84.2_{(\pm 15)}$ | $0.9195_{(\pm 0.006)}$ | $777.9_{(\pm 102)}$ | $99.48_{(\pm 5.6)}$ |
| NeurSR | $0.5054_{(\pm 0.058)}$ | $17.47_{(\pm 1.6)}$ | $12.74_{(\pm 0.37)}$ | $0.3823_{(\pm 0.020)}$ | $25.81_{(\pm 1.7)}$ | $13.54_{(\pm 0.16)}$ |
| E2ESR | $0.5152_{(\pm 0.038)}$ | $5.621_{(\pm 4.9)}$ | $38.49_{(\pm 2.7)}$ | $0.7714_{(\pm 0.014)}$ | $6.183_{(\pm 0.83)}$ | $44.09_{(\pm 0.61)}$ |
| SNIP | $0.9187_{(\pm 0.042)}$ | $5650.7_{(\pm 35)}$ | $39.35_{(\pm 2.3)}$ | $0.9917_{(\pm 0.020)}$ | $6514.8_{(\pm 821)}$ | $37.71_{(\pm 2.4)}$ |
| GPlearn | $0.8228_{(\pm 0.052)}$ | $894.6_{(\pm 108)}$ | $25.84_{(\pm 2.9)}$ | $0.8911_{(\pm 0.007)}$ | $2938_{(\pm 543)}$ | $48.83_{(\pm 9.5)}$ |
| AFP | $0.9110_{(\pm 0.065)}$ | $161.4_{(\pm 28)}$ | $44.44_{(\pm 5.3)}$ | $0.9577_{(\pm 0.007)}$ | $3886_{(\pm 341)}$ | $40.79_{(\pm 1.5)}$ |
| AFP-FE | $0.9496_{(\pm 0.037)}$ | $10082_{(\pm 565)}$ | $50.96_{(\pm 4)}$ | $0.9826_{(\pm 0.005)}$ | $28812_{(\pm 0.51)}$ | $48.87_{(\pm 1.4)}$ |
| EPLEX | $0.8822_{(\pm 0.078)}$ | $405.8_{(\pm 43)}$ | $53.46_{(\pm 2.3)}$ | $0.9901_{(\pm 0.004)}$ | $10283_{(\pm 532)}$ | $45.62_{(\pm 1.3)}$ |
| SBP-GP | $0.9323_{(\pm 0.050)}$ | $21886_{(\pm 782)}$ | $901.2_{(\pm 34)}$ | $0.9905_{(\pm 0.007)}$ | $28932_{(\pm 83)}$ | $621.9_{(\pm 6.6)}$ |
| GP-GOMEA | $0.9668_{(\pm 0.031)}$ | $402.9_{(\pm 802)}$ | $43.71_{(\pm 2.1)}$ | $0.9957_{(\pm 0.003)}$ | $3186_{(\pm 454)}$ | $46.43_{(\pm 0.91)}$ |
| Operon | $0.9380_{(\pm 0.042)}$ | $97.13_{(\pm 15)}$ | $83.44_{(\pm 1.2)}$ | $0.9847_{(\pm 0.008)}$ | $3090_{(\pm 330)}$ | $89.23_{(\pm 0.65)}$ |
| SPL | $0.7715_{(\pm 0.044)}$ | $355.3_{(\pm 59)}$ | $13.91_{(\pm 1.6)}$ | $0.7109_{(\pm 0.013)}$ | $270_{(\pm 168)}$ | $13.54_{(\pm 0.50)}$ |
| DSR | $0.8086_{(\pm 0.047)}$ | $500_{(\pm 300)}$ | $18.51_{(\pm 2.2)}$ | $0.8779_{(\pm 0.002)}$ | $814.9_{(\pm 197)}$ | $16.03_{(\pm 0.46)}$ |
| RSRM | $0.5553_{(\pm 0.057)}$ | $138.3_{(\pm 6.4)}$ | $13.54_{(\pm 1.1)}$ | $0.8104_{(\pm 0.016)}$ | $133.8_{(\pm 6)}$ | $12.81_{(\pm 0.41)}$ |
| AIFeynman2 | $0.3170_{(\pm 0.082)}$ | $65.97_{(\pm 19)}$ | $23.53_{(\pm 4.6)}$ | $0.2248_{(\pm nan)}$ | $710.7_{(\pm nan)}$ | $176.6_{(\pm nan)}$ |
| BSR | $0.7190_{(\pm 0.076)}$ | $23292_{(\pm 510)}$ | $49.54_{(\pm 4.9)}$ | $0.6567_{(\pm 0.024)}$ | $32497_{(\pm 7914)}$ | $28.77_{(\pm 1.1)}$ |
| MDL | $0.9686_{(\pm 0.057)}$ | $522_{(\pm 43)}$ | $20.5_{(\pm 0.62)}$ | $0.9097_{(\pm 0.009)}$ | $962.9_{(\pm 918)}$ | $30.86_{(\pm 1.4)}$ |
| TPSR | $0.9707_{(\pm 0.022)}$ | $159.1_{(\pm 23)}$ | $56.28_{(\pm 4.1)}$ | $0.9836_{(\pm 0.005)}$ | $184.01_{(\pm 8.3)}$ | $66.96_{(\pm 1.9)}$ |
| RAG-SR | $0.9693_{(\pm 0.043)}$ | $385.3_{(\pm 37)}$ | $46.23_{(\pm 2.7)}$ | $0.9849_{(\pm 0.006)}$ | $2957_{(\pm 306)}$ | $75.12_{(\pm 2.6)}$ |
| Ours | $0.9773_{(\pm 0.029)}$ | $257.64_{(\pm 11)}$ | $20.21_{(\pm 1.3)}$ | $0.9886_{(\pm 0.005)}$ | $618.9_{(\pm 64)}$ | $23.62_{(\pm 0.8)}$ |

# H COMPARISON WITH THE STANDARD BAYESIAN OPTIMIZATION (BO) ALGORITHM ON FEYNMAN DATASET

Table 8: Comparison of CMA and standard BO under different $k$ with 512-dimensional latent space

|  | Method | $R^2\uparrow$ | Time (s)$\downarrow$ | Complx$\downarrow$ |
|---|---|---|---|---|
| k=256 | CMA | 0.9872 | 192.3 | 22.94 |
| | standard BO | 0.9274 | 295.8 | 21.37 |
| k=128 | CMA | 0.9675 | 179.1 | 23.26 |
| | standard BO | 0.9161 | 237.5 | 22.55 |
| k=64 | CMA | 0.9388 | 173.2 | 23.93 |
| | standard BO | 0.8974 | 165.6 | 23.07 |

Table 8 reports the comparison between CMA-ES and standard Bayesian Optimization (BO) on GenSR with the 512-dimensional latent space. In our main experiments, GenSR adopts $k = 256$, corresponding to the top-$k$ principal latent dimensions with the largest variance. We therefore first conduct BO experiments under this same setting. As shown in the table, BO attains slightly lower expression complexity than CMA-ES, but its performance in terms of accuracy ($R^2$) and optimization time is substantially worse. This may be attributed to the challenges that BO potentially faces when dealing with high-dimensional optimization problems.

To further investigate the effect of reducing the number of optimized dimensions, we additionally evaluate $k = 128$ and $k = 64$ (a setting already examined for CMA-ES in the $k$ ablation study reported in Appendix 7). Although decreasing $k$ improves the runtime of BO, its accuracy does not increase accordingly. At the same time, the accuracy of CMA-ES drops significantly under smaller $k$ values. These results suggest that overly small $k$ leads to an excessive loss of important latent dimensions, thereby limiting the expressiveness of the search space and degrading the performance of both optimization methods.

Table 9: Comparison of CMA and standard BO under different $k$ with 256-dimensional latent space

|  | Method | $R^2\uparrow$ | Time (s)$\downarrow$ | Complx$\downarrow$ |
|---|---|---|---|---|
| k=256 | CMA | 0.6984 | 181.4 | 18.23 |
| | standard BO | 0.6241 | 206.6 | 17.12 |
| k=128 | CMA | 0.4935 | 174.5 | 19.08 |
| | standard BO | 0.4797 | 178.3 | 18.67 |

We further evaluate another pretrained model used in Appendix E.1, namely the GenSR variant with the 256-dimensional latent space, and conduct experiments with $k = 256$ and $k = 128$. As shown in Table 9, both CMA-ES and standard BO exhibit relatively low accuracy under this lower-dimensional latent representation, which may suggest that the expressive capacity of the latent space is notably reduced in such settings. Moreover, the performance gap between BO and CMA-ES becomes less pronounced.

Overall, these observations imply that BO may face certain limitations when dealing with high-dimensional optimization, whereas CMA-ES appears relatively more robust for our task. Attempts to improve BO by reducing the latent dimensionality or decreasing the number of optimized dimensions may still leave accuracy constrained, potentially due to insufficient latent expressiveness or the omission of important dimensions.

# I ABLATION ANALYSIS OF THE POSTERIOR SYMBOLIC ENCODER BRANCH

To examine the role of the dual-branch framework in shaping the latent space, we construct a variant that explicitly disrupt the posterior branch, denoted as **GenSR w/o Posterior**. While keeping the

Table 10: Comparison between GenSR and its disrupted variant

| Method | $R^2\uparrow$ | Complx$\downarrow$ |
|---|---|---|
| GenSR | 0.921 | 19.04 |
| GenSR w/o Posterior | 0.752 | 26.41 |

overall two-branch architecture intact, we pair numerical samples $X$ with randomly sampled equations $F$ as inputs, and replace the reconstruction loss of the posterior branch with a cross-entropy loss computed against the target expression corresponding to $X$. In addition, to directly evaluate the inherent performance of the model under this setting, we remove the CMA-ES optimization step and rely solely on the initial localization produced by the pretrained model.

As shown in Table I, disabling the posterior branch leads to a substantial drop in performance ($R^2$ decreases from 0.921 to 0.752, and the complexity notably increases), indicating that the underlying latent-space structure is significantly disrupted. Consequently, the initial localization becomes unreliable, demonstrating the critical role of the dual-branch framework in maintaining a well-structured latent representation and overall model performance.

## J COMPARISON WITH THE STANDARD BAYESIAN OPTIMIZATION (BO) ALGORITHM

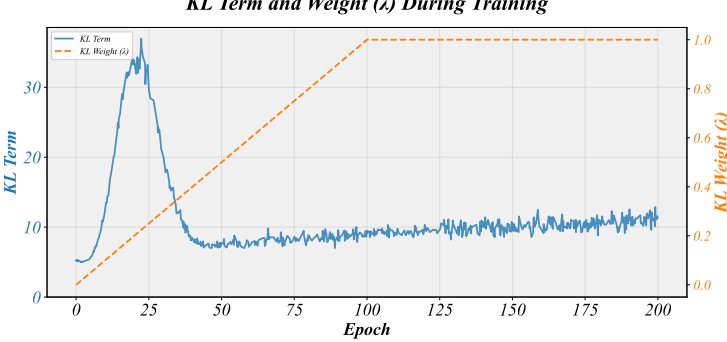

Figure 10: The horizontal axis denotes the epoch, while the left and right vertical axes correspond to the KL term and the KL weight, respectively. The KL weight annealing strategy used here increases the weight from 0 to 1.0 during the first 50% of the training epochs and keeps it constant thereafter.

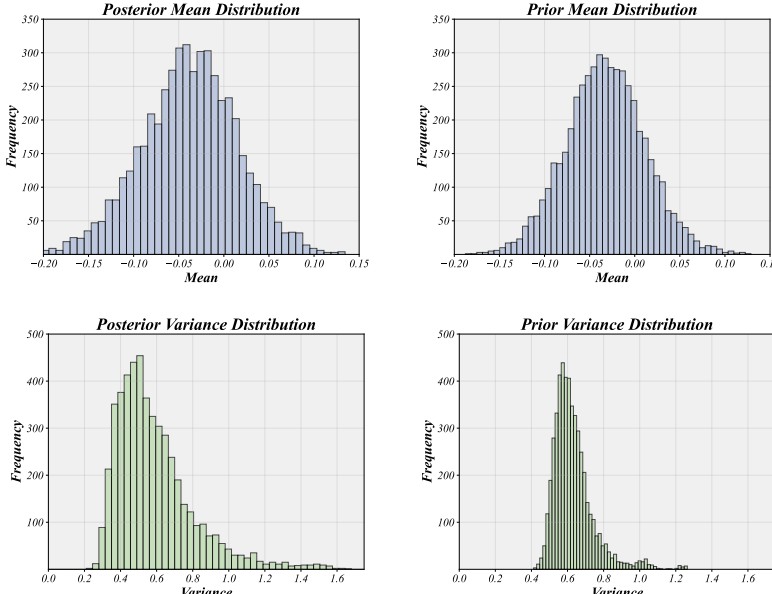

Figure 11: Estimated distributions of prior and posterior means and variances obtained from 5,000 randomly sampled inputs in GenSR.

As described in Appendix F.2, we adopt a KL weight annealing strategy during training. As shown in Fig. 10, the KL weight gradually increases from 0 to 1 in the first half of training, while the KL term eventually settles into a relatively stable range in the later stages, indicating that no collapse occurs.

In addition, in GenSR, the prior distribution is learnable rather than fixed to a standard Gaussian. During training, the continual updates of the prior prevent the posterior distribution from trivially collapsing toward it, which may help reduce the likelihood of posterior collapse. Fig. 11 shows the distributions of the prior and posterior means and variances computed from 5,000 randomly generated samples using the final pretrained model. Clear discrepancies are observed in both the mean and variance distributions, indicating that the prior and posterior branches have not collapsed into a single distribution. If posterior collapse had occurred, the posterior would converge toward the prior; however, the differences shown here demonstrate that the model maintains a non-degenerate and informative posterior.

## K    COMPARING GENERATIVE AND DISCRIMINATIVE LATENT SPACES

Contrastive learning methods (e.g., SNIP) construct a discriminative latent space that is fragmented and contains numerous "holes," such that latent vectors sampled between two valid points often decode to empty or otherwise invalid strings. In contrast, our GenSR constructs a generative space (a continuous manifold). It learns the distribution of equations — meaning we can sample continuously and decode valid equations. This allows for "smooth interpolation," which is a prerequisite for the fine-grained search (CMA-ES) we perform. Table K provides a clear summary of the differences between the generative and discriminative latent spaces. Besides, the interpolation experiments (Appendix C.2) confirm that SNIP often falls into semantically invalid areas during search.

Table 11: Comparison between **Generative latent space** and **Discriminative latent space**

| Property | Generative latent space (GenSR) | Discriminative latent space (contrastive) |
| --- | --- | --- |
| Learning Objective | Equation distribution | Pointwise alignment |
| Continuous latent sampling | Supported | Not supported |
| Suitability for search | High | Low: limited with invalid regions |

