# OpenReview forum: "GenSR: Symbolic regression based on equation generative space"
_ICLR.cc/2026/Conference — ICLR 2026 Poster_

### Official Review · Reviewer_A4iV · 2025-10-27

**Soundness:** 3
**Presentation:** 3
**Contribution:** 2
**Rating:** 6
**Confidence:** 3

**Summary:**

This paper proposes GenSR, a novel symbolic regression framework that transforms the discrete equation search space into a continuous generative latent space using a dual-branch conditional variational autoencoder. The method follows a "map construction → coarse localization → fine search" paradigm, where the CVAE learns a latent space with global symbolic continuity and local numerical smoothness. At inference, a modified CMA-ES algorithm refines candidate equations in this latent space. The authors provide a Bayesian interpretation and demonstrate strong empirical performance on SRBench datasets.

**Strengths:**

1. This paper presents a highly promising idea by formulating the symbolic regression problem as an optimization task in a continuous latent space. This approach transforms the search space from a structured space and allows continuous optimization algorithms such as CMA-ES to be applied directly. While prior work like SNIP has explored latent space optimization, constructing a smooth and meaningful latent space remains an open question because it is crucial for effective latent space optimization.
2. The authors propose a dual-branch conditional variational autoencoder to decouple and align symbolic and numerical information. This design is clearly motivated. Using KL divergence loss to align the prior and posterior distributions is a reasonable approach to promote local numerical smoothness.
3. The experimental evaluation is thorough. The authors test on the SRBench benchmark and compare GenSR with 18 baseline methods. In addition to main results, the paper reports noise robustness, latent space t-SNE visualizations, interpolation experiments, and a detailed ablation study. Figures 4 and 5 demonstrate that GenSR achieves better structural disentanglement and numerical continuity in the latent space than competing approaches.

**Weaknesses:**

1. The architectural choice of using a VAE appears somewhat outdated, as the overall model seems to be a slightly modified version of E2ESR, with an additional layer appended to the encoder’s final hidden state to map to ($\mu$, $\sigma$). A well-known issue with VAEs is posterior collapse, where the latent variables become uninformative and the decoder effectively ignores them (this problem can be even more pronounced with high-capacity models such as transformers). It would be beneficial to provide empirical evidence (KL divergence during training) to demonstrate that posterior collapse is indeed being avoided.
2. Although Section 3.3 provides an explanation for the desired symbolic continuity and numerical smoothness, its reasoning remains informal. For example, the paper assumes that the overlap of high-probability regions for similar equations enables smooth interpolation (lines 203–218). However, this is merely asserted with no empirical study of how frequently this occurs in practice. Similarly, the connection between minimizing the KL divergence and inducing local numerical smoothness (lines 219–234) is also asserted rather than formally established.

**Questions:**

1. The interpolation experiment in Appendix C.2 is vague. Linear interpolation in high-dimensional spaces can be problematic. For a VAE with a standard Gaussian prior, most of the probability mass is concentrated in a thin shell far from the origin. Linear interpolation between two points is highly likely to traverse low-probability regions of the latent space.  Could the authors clarify the specific setup for this experiment?
2. The paper reformulates the task as a Bayesian optimization problem. Have the authors considered applying Bayesian optimization algorithms directly in the latent space? I would be interested to see a comparison between CMA-ES and standard BO methods in terms of sample efficiency and the quality of the final solutions.

Please note that if the authors are able to address my questions during the rebuttal and discussion phase, I am willing to increase my score and confidence.

---

> ### Author Response · Authors · 2025-11-22
> **Response to the Reviewer A4iV (part 1)**
>
> Thank you very much for your careful and insightful feedback. Your comments provide us with valuable perspectives and significantly help us refine our work. Below we address your questions and concerns point by point.
> ___
> * **Reason of choosing the CVAE architecture (W1)**
>
> We understand the reviewer’s concern on the novelty of architecture. However, we would like to emphasize that **our dual-branch CVAE design is driven jointly by theoretical derivation and clear motivation**.
>
> **1. Theoretical Perspective:** We formulate SR as maximizing the conditional probability p(Equ.|Num.), and derive its ELBO, which directly corresponds to our dual-branch CVAE framework. To the best of our knowledge, no existing work has approached SR from the perspective of maximizing the ELBO of p(Equ.|Num.). Therefore, We believe this probabilistic formulation and the resulting architecture offer unique theoretical value.
>
> **2. Motivational Perspective:** To construct a well-organized generative latent space, we do not adopt the standard CVAE with a fixed Gaussian prior. Instead, we design a learnable dual-branch CVAE framework, and model coupled symbolic–numerical smoothness by symbolic reconstruction and distribution alignment. To further demonstrate the necessity of our dual-branch CVAE, we followed Reviewer CAH6's suggestion and add an ablation study in Appendix I that disrupts the dual-branch structure. The results show in Appendix I the critical role of our architecture in constructing an effective latent space.
>
> **3. Comparison with E2E:** Although both E2E and GenSR use transformer-based architectures, E2E does not construct a generative latent space for equations (Appendix K further explains why such a space is crucial). As a result, E2E neither supports latent-space search nor accounts for the essential properties an equation latent space should satisfy. Consequently, GenSR achieves a better balance among accuracy, expression complexity, and efficiency than E2E.
>
> Thank you for the feedback. We will consider more advanced architectures that are well suited for SR tasks in future work.
> ___
> * **Regarding Posterior Collapse (W1)**
>
> We sincerely thank the reviewer for raising this important concern. We provide additional demonstrations (more details please refer to Appendix J): Appendix J shows the KL divergence trend across epochs under KL-weight annealing. We also visualize the prior and posterior distributions over 5000 input pairs and observe that their behaviors are different. These observations indicate that posterior collapse does not occur in our training. We attribute this to two main factors:
>
> **1. KL annealing:** As stated in lines 968 and 982, we employed KL annealing to stabilize the training process. KL annealing gradually increases the KL weight, limiting the decoder capacity and forcing the model to rely on latent variables. We have added an additional clarification in the main text of revision (line 201).
>
> **2. Learnable prior:** Our prior is learnable rather than fixed. During early training, the evolving prior prevents the posterior from easily collapsing onto it, reducing the risk of posterior collapse.
> ___
>
> * **Additional Explanation on Latent-Space Smoothness (W2)**
>
> We acknowledge that Sec. 3.3 provides a formal analysis rather than a rigorous mathematical proof. Actually, many prior works have demonstrated that VAE latent spaces naturally exhibit strong continuous and interpolable structure [1–3]. Nonetheless, we introduce a multi-sampling strategy combined with a feature-fusion module, which significantly increase the overlap of high-probability regions for similar expressions, further enhancing symbolic continuity (we add explanation in line 217-220 of the revision). Table 1 provides an ablation study, which shows that our strategy plays a key role in constructing a higher-quality latent space.
>
> &nbsp;&nbsp;&nbsp;&nbsp; **Table 1.** Ablation study of multi-sampling strategy
>
> | Metric                     | Single sampling (copy n times)| Sample n times (n=200) |
> |---------------------------|-----------------|--------------------------|
> | R^2 (without CMA-ES)      | 0.846           | 0.921                    |
> |
>
> For numerical smoothness, our analysis follows SNIP (Sec. 5.2 of SNIP). By aligning the distributions of the two modalities—rather than enforcing pointwise similarity as in SNIP—we obtain stronger local numerical smoothness. Since numerical smoothness does not benefit from a global mechanism, and based on our visualizations, we cautiously characterize this smoothness as local rather than global.
> ___

---

> ### Author Response · Authors · 2025-11-22
> **Response to the Reviewer A4iV (part 2)**
>
> * **Clarification of the Interpolation Experiment (Q1)**
>
> We appreciate your feedback and have provided detailed clarifications in Appendix C.2 of the revision. Briefly, we compute latent encodings $z_1$, $z_2$ for two expressions via GenSR’s prior branch. The same procedure is applied to E2ESR and SNIP by their numerical encoder. We then perform linear interpolation $z(ratio)=(1-ratio)z_1+ratio\,z_2$ with $ratio\in \{ 0,0.25,0.5,0.75,1 \}$, and decode the $z(ratio)$.
>
> **1. Explanation of linear interpolation in high-dimensional spaces:** Your viewpoint is insightful. However, linear interpolation of latent vectors remains a common experiment for evaluating interpolability of VAE-based methods, such as [4-6]. In our opinion, while the KL term in standard VAEs pushes the posterior toward a standard Gaussian (the “thin shell”), the reconstruction term plays a crucial counter-role. It prevents the latent distribution to collapse to a standard Gaussian and enhances decoding validity in lower-probability regions (low-probability regions may be affected by the reconstruction loss of the sampled points, since the sampling process cannot guarantee that the sampled points always lie in high-probability regions). Consequently, linear interpolation has consistently yielded good results in prior VAE-based works [4-6].
> For our GenSR specifically, we avoided posterior collapse by using annealing and a learnable prior, and our multi-sampling strategy may further ensure that low-probability regions remain valid for decoding.
>
>
> **2. More Discussion:** We find that interpolation quality depends on the expressive power of the latent space. When the latent space captures the data manifold well, interpolation is reliable (e.g., results on MNIST [5]); however, when it lacks the capacity to represent complex data, interpolation quality degrades (e.g., results on high-fidelity faces [6]). We then evaluate interpolation using the 256-dimensional pretrained model (from Appendix E.1) below. Although no “Invalid” results occurred, the transitions were obviously less smooth than our results in Appendix C.2 (512-dimensional space).
>
> | ratio | Results |
> |-------|-------|
> | **Pair 1:** Expression 1: *c exp(x) · sin(x − c)* → Expression 2: *log(x + c) · cos(x)* |
> | 0     | c exp(x) sin(x − c) |
> | 0.25  | c exp(x) log(x − c) - cos(x) sin(cx) |
> | 0.5   | exp(x) log(x − c) + cos(x) sin(cx) |
> | 0.75  | log(x + c) cos(x) |
> | 1     | log(x + c) cos(x) |
> |
> ___
>
> * **Comparison with BO (Q2)**
>
> We greatly appreciate this suggestion and have added a comparison with standard BO in Appendix H of revision. We evaluate multiple latent-space dimensions (from Appendix E.1) and multiple top-k update settings (as in Section 3.4).
>
> **1. Results on the main model (512-dimensional latent space):** As shown in Table 2, with k=256, BO yields slightly simpler expressions but is significantly worse than CMA-ES in both accuracy and search efficiency. Reducing k improves BO efficiency but does not improve accuracy. CMA-ES also degrades sharply because too many latent dimensions are ignored.
>
> &nbsp;&nbsp;&nbsp;&nbsp; **Table 2** Comparison of CMA and standard BO under different k with 512-dimensional latent space
>
> | k value | Method      | R<sup>2</sup> | Time (s) | Complx |
> |---------|-------------|---------------|----------|--------|
> | **k = 256** | CMA         | 0.9872        | 192.3    | 22.94  |
> |         | standard BO | 0.9274        | 295.8    | 21.37  |
> | **k = 128** | CMA         | 0.9675        | 179.1    | 23.26  |
> |         | standard BO | 0.9161        | 237.5    | 22.55  |
> | **k = 64**  | CMA         | 0.9388        | 173.2    | 23.93  |
> |         | standard BO | 0.8974        | 165.6    | 23.07  |
> |
>
> **2. Results on the low-dimensional model (256-dimensional latent space):** The results in Table 3 show that both BO and CMA-ES suffer substantial performance drops due to insufficient latent representational capacity.
>
> &nbsp;&nbsp;&nbsp;&nbsp;  **Table 3** Comparison of CMA and standard BO under different k with 256-dimensional latent space
>
> | k value | Method       | R<sup>2</sup> | Time (s) | Complx |
> |---------|--------------|---------------|----------|--------|
> | **k = 256** | CMA          | 0.6984        | 181.4    | 18.23  |
> |         | standard BO  | 0.6241        | 206.6    | 17.12  |
> | **k = 128** | CMA          | 0.4935        | 174.5    | 19.08  |
> |         | standard BO  | 0.4797        | 178.3    | 18.67  |
> |
>
> Overall, it seems that BO is not well-suited for optimizing in such high-dimensional generative latent spaces. Reducing latent dimensionality or restricting updating dimensions improves BO efficiency but severely harms expression quality.
> ___

---

> ### Author Response · Authors · 2025-11-22
> **Response to the Reviewer A4iV (part 3)**
>
> * **Final Remark**
> Thank you once again for the remarkably thoughtful and inspiring feedback. Many of your suggestions have meaningfully shaped our revision, and it was genuinely a pleasure to incorporate them. We are very happy with how much your insights helped improve the manuscript, and we would be delighted to continue refining the work if you have any additional thoughts or suggestions during the discussion phase.
> ___
> [1]Lyu Z, Ali S, Breazeal C. Introducing variational autoencoders to high school students, Proceedings of the AAAI conference on artificial intelligence. 2022, 36(11): 12801-12809.
> [2]Gómez-Bombarelli R, Wei J N, Duvenaud D, et al. Automatic chemical design using a data-driven continuous representation of molecules[J]. ACS central science, 2018, 4(2): 268-276.
> [3] https://www.codecademy.com/article/variational-autoencoder-tutorial-vaes-explained.
> [4] Zhang J, Bai J, Lin C, et al. Improving variational autoencoders with density gap-based regularization[J]. NIPS, 2022, 35: 19470-19483 (Supplementary Material, Appendix F).
> [5]Michelis M Y, Becker Q. On Linear Interpolation in the Latent Space of Deep Generative Models[C]//ICLR 2021 Workshop on Geometrical and Topological Representation Learning.
> [6]Casale F P, Dalca A, Saglietti L, et al. Gaussian process prior variational autoencoders[J]. Advances in neural information processing systems, 2018, 31.

---

> ### Author Response · Authors · 2025-11-27
> **Looking forward to discussion**
>
> Dear Reviewer A4iV,
>
> We hope our answers and new experiments have addressed your concerns and questions. Please let us know if you have any more questions before the end of the discussion period.
>
> Thanks for your dedicated service to the community！

---

> ### Author Response · Authors · 2025-11-28
> **Response to the Reviewer A4iV**
>
> We sincerely thank the reviewer for the further comments and the strong support for our work.
>
> >  **Further comments on Q2 & W2:** *“Q2: There has been progress in performing Bayesian optimization in latent spaces, including for symbolic regression [^1] and molecular optimization tasks[^6]. Although optimization is not the main contribution of this paper, I believe that performing optimization in latent space is as important as constructing the latent space. Compared with genetic programming and end-to-end symbolic expression generation, the symbolic regression task here is decomposed into two steps: (1) constructing a latent space for symbolic expressions; (2) performing optimization in that latent space. Recent studies have shown that effective exploration in latent space is challenging, with difficulties including but not limited to: how to avoid exploring low-probability regions[^2][^3][^4], and the non-smoothness of the latent space[^5], etc. From the authors’ experimental results, vanilla BO performs relatively poorly, which also suggests that directly applying BO algorithms to high-dimensional optimization remains challenging. In addition, constraining the covariance matrix of CMA-ES to be diagonal in the paper is very similar to Sep-CMA-ES[^8], and should be properly cited if necessary. W2: I believe there exists some metrics that may help quantify the smoothness of the latent space. For example, the gradient norm of decoded samples[^7]. As in CoBO[^5], smoothness can be indirectly characterized via the correlation between latent-space distance and the difference in target values.”*
>
> Thank you for your suggestions and the valuable references you provided. We fully agree with your insight that "performing optimization in latent space is as important as constructing the latent space." In our future work, we will explore more optimization algorithms in the latent space, as we believe that CMA-ES is not the only viable option. Besides, thank you very much for the reminder, we will cite Sep-CMA-ES in the revision.
>
> Thank you for your suggestion regarding the use of the gradient norm. In our future work, we will still be committed to exploring latent spaces that are specifically tailored for SR tasks. We hope that GenSR can serve as an inspiration for future research.
>
> >  **About Reviewer mtFG and score upgrade** *"I am satisfied with the quality of the paper and the authors’ response. Compared with existing methods based on genetic programming and end-to-end generation, this work appears to be a good baseline for symbolic regression in latent space. Although one reviewer strongly opposed the paper with the highest confidence score before the deadline, I still believe that the contributions of this paper are different from prior work. Due to data leakage issues, it currently seems that reviewers are not allowed to adjust their scores. If the system later re-enables score modification, I will raise my score from weak accept to accept and increase my confidence."*
>
> We sincerely thank you for your strong support and recognition of our work. Regrettably, due to the latest ICLR announcement, we may not be able to officially receive your score update. However, your recognition serves as a significant encouragement for our future research.
>
> >**About the source code** *"Given that there is currently no anonymous code repository for this work, I suggest that, if the paper is accepted, the authors release their implementation as soon as possible."*
>
> We are fully committed to reproducibility. We will release the source code, pre-trained model weights, and checkpoints upon acceptance. We look forward to your further consideration.

---

### Official Review · Reviewer_CAH6 · 2025-10-28

**Soundness:** 4
**Presentation:** 4
**Contribution:** 4
**Rating:** 8
**Confidence:** 5

**Summary:**

The paper proposes GenSR, a Conditional-VAE-based symbolic regression model that encodes equations into a continuous latent space and performs CMA-ES optimization there. It claims improved performance over existing symbolic methods such as E2ESR and SNIP.

**Strengths:**

A key strength of the paper is the clear comparison with SNIP, which highlights the limitations of contrastive learning in SNIP and demonstrates the advantage of using a VAE framework for symbolic regression. Although using autoencoders for symbolic regression has been explored in the genetic programming literature [1], this paper presents a meaningful improvement by extending the autoencoder to a pre-training scenario and showing the benefits of a conditional VAE design.

[1]. Wittenberg, David, Franz Rothlauf, and Christian Gagné. "Denoising autoencoder genetic programming: strategies to control exploration and exploitation in search." Genetic Programming and Evolvable Machines 24.2 (2023): 17.

**Weaknesses:**

The main weakness is the lack of an ablation study to justify the necessity of the posterior symbolic encoder branch.

**Questions:**

1. The paper should clarify why encoding the symbolic expression is required. In a CVAE, the encoder is necessary because many outputs can correspond to the same condition, such as images. However, in symbolic regression, for a given dataset, there are usually only a few valid equations. Please consider adding an ablation where the symbolic encoder/posterior branch is removed, to test whether the bottleneck itself is the key contributor to performance.

2. Figure 5 illustrates numerical continuity, but the paper does not explain how this continuity emerges, since no explicit loss term enforces it. Please provide an explanation of this phenomenon. While it is expected that a VAE can cluster similar functions, the observed continuity is an interesting behavior that deserves discussion.

3. On page 21, one citation appears incorrect and is displayed as a question mark. Please correct this.

4. The model weights and source code should be released for reproducibility.

5. Some methods listed in Table 4 are not reported in Table 3. Please include their performance on the black-box dataset for completeness.

---

> ### Author Response · Authors · 2025-11-22
> **Response to the Reviewer CAH6**
>
> We appreciate your positive rating and the insightful comments, particularly regarding the ablation study on the posterior branch and the mechanism behind numerical continuity. Below, we provide clarifications and new experimental results to address your questions.
> ___
> * **Necessity of the Posterior Branch and Ablation Study (W1 & Q1)**
>
> We appreciate this valuable suggestion. The design of the posterior branch is driven by both motivational and theoretical considerations.
>
> **1. Motivational Perspective:** The main role of posterior branch is to construct a well-structured space where symbolic patterns remain separable (symbolic continuity). As you noted, “VAE can cluster similar functions” and this effect is precisely achieved by the posterior branch’s reconstruction loss. Symbolic continuity enables the model to produce an initial localization similar with the target equation’s symbolic structure, offering a better starting point for fine-grained search.
>
> **2. Theoretical Perspective:** Training GenSR is equivalent to maximizing the ELBO of the conditional distribution p(Equ.|Num.). The posterior encoder approximates the variational distribution, while the decoder models the conditional likelihood. The reconstruction loss of the posterior branch corresponds directly to the first term of the ELBO. Therefore, from a Bayesian inference perspective, the posterior branch is an indispensable component of our framework.
>
> **3. Ablation Study:** Thanks for your valuable suggestion. We have added an ablation study to verify the necessity of the posterior branch in Appendix I. To disable the posterior branch, we paired numerical samples $X$ with random equations $F$ as inputs and replaced the posterior reconstruction loss with a cross-entropy loss against the target expression of $X$. We then evaluated the initial accuracy (without CMA-ES). As shown in Table 1, disabling the posterior branch leads to a severe drop in performance. This confirms that the posterior branch is critical for maintaining a well-structured latent space and ensuring effective initial localization.
>
> &nbsp;&nbsp;&nbsp;&nbsp; **Table 1**: Comparison between GenSR and its disrupted variant
>
> | Method             | R^2 (without CMA-ES)  | Complexity  |
> |--------------------|--------|--------------|
> | GenSR              | 0.921  | 19.04        |
> | GenSR w/o Posterior| 0.752  | 26.41        |
> |
>
> ___
> * **Explanation of Local Numerical Continuity (Q2)**
>
> We are happy to address your questions. As analyzed in Sec. 3.3, **numerical continuity arises from the learnable prior branch and the KL-divergence loss**. The prior branch models the numerical feature distribution and aligns it with the symbolic distribution learned by the posterior branch via the KL loss. This alignment is conceptually similar to SNIP, where contrastive learning aligns embeddings across two modalities. Consequently, like Sec. 5.2 of SNIP, we observe numerical smoothness in the latent space. However, there is a key difference: contrastive learning in SNIP aligns point-to-point embeddings, whereas our KL divergence aligns the distributions of two modalities. This leads to a smoother and better organized latent space, resulting in more efficient search and higher accuracy than SNIP.
>
> ___
>
> * **Clarifications on Other Questions (Q3, Q4, Q5)**
>
> 1. Q3 (Typo): Thank you for spotting this. We have corrected it in the revision.
> 2. Q4 (Reproducibility): We are fully committed to reproducibility. We will release the source code, pre-trained model weights, and checkpoints upon acceptance.
> 3. Q5 (Missing Baselines): Thank you for spotting this. We followed the experimental settings of MDLFormer (ICLR 2025), which led to this omission. We have added the results of these methods on the black-box dataset in the revision.
> ___
> * **Final remark**
>
> We strictly appreciate your strong support and the score of 8 and confident level 5. Your feedback has been invaluable in strengthening the manuscript, particularly regarding the suggestion of posterior branch ablation study. We sincerely welcome any further suggestions.

---

> ### Author Response · Authors · 2025-11-27
> **Looking forward to discussion**
>
> Dear Reviewer CAH6,
>
> We hope our answers and new experiments have addressed your concerns and questions. Please let us know if you have any more questions before the end of the discussion period.
>
> Thanks for your dedicated service to the community！

---

### Official Review · Reviewer_sUgp · 2025-10-31

**Soundness:** 3
**Presentation:** 4
**Contribution:** 3
**Rating:** 6
**Confidence:** 3

**Summary:**

This paper points out that "similar structure ≠ similar value" in discrete symbol space leads to noisy fitting error signal and lack of search direction, which is the source of inefficiency of most SR methods. Therefore, GenSR is proposed. The pre-trained dual-branch CVAE reparameterizes the equation into a generative latent space with "global symbolic continuity + local numerical smoothness", according to the "map construction → rough positioning → fine search" paradigm and improves CMA-ES search. SR is reformulated as a Bayesian problem and implemented by ELBO training.

**Strengths:**

1. This method seems to achieve very good experimental results
2. This paper experiment is relatively complete, considerable workload

3. In addition, the authors give a Bayesian perspective and reformulate SR as a probabilistic inference problem of maximizing P(F|X) trained by ELBO, which provides a clear theoretical framework and interpretability support for the rationality of the method.

**Weaknesses:**

The article is well written, but it seems to be only transferring the method in the multi-modal of graphics and text to the field of symbolic regression, and the innovation of the algorithm is not very high. Ask the author to emphasize his technical innovation

**Questions:**

1. What are the advantages of methods trained via CVAE over methods trained via contrastive learning?
2. What's the reason? It seems that contrastive learning can achieve similar results to the map building mentioned in this article, and I think it might even be better.

---

> ### Author Response · Authors · 2025-11-22
> **Response to the Reviewer sUgp (part 1)**
>
> We sincerely thank the reviewer for the positive comments of our writing quality, experimental depth, and theoretical clarity. We appreciate the opportunity to clarify the technical innovations of GenSR and to discuss the critical differences between our generative approach and contrastive learning methods.
> ___
> * **Clarification on technical innovation (Weakness)**
>
> We understand the reviewer’s concerns about technical innovation. While GenSR leverages components like CVAEs and CMA-ES, it is not a direct transfer of multi-modal methods to SR. Instead, GenSR is driven by task-specific motivation and theoretical grounding to address the unique smoothness of the equation space. Just as SNIP (NIPS 2023) adapted CLIP-like structures to SR with novel contributions, GenSR adapts CVAEs and CMA-ES with the following specific innovations:
>
> **1. Dual-branch CVAE:** Traditional CVAE-based frameworks use fixed standard Gaussian prior, which fails to couple symbolic structures with numerical patterns effectively. GenSR introduces a dual-branch CVAE with a learnable numerical prior and a symbolic posterior, enabling a generative latent space that is smooth in both modalities. As Reviewer A4iV noted, “This design is clearly motivated”.
> We add a new ablation study that disrupt the dual-branch CVAE to assess its impact on the latent space (see Appendix I for details). The results in Table 1 of the initial accuracy show a sharp degradation in latent-space quality once the dual-branch CVAE is disrupted, confirming the necessity of the design.
>
> &nbsp;&nbsp;&nbsp;&nbsp;**Table 1**: Comparison between GenSR and its disrupted variant
>
> | Method             | R^2 (without CMA-ES)    | Complexity  |
> |--------------------|--------|--------------|
> | GenSR              | 0.921  | 19.04        |
> | GenSR w/o Posterior| 0.752  | 26.41        |
> |
>
> **2. Multi-sampling + Feature Fusion:** Unlike standard VAEs that rely on single sample estimates, we introduce a multi-sampling strategy combined with a feature-fusion module. This is tailored to the SR task to enhance latent-space smoothness. We highlight this design in the revision (lines 217–220) and add ablation in Table 2, confirming effectiveness of this strategy.
>
> &nbsp;&nbsp;&nbsp;&nbsp; **Table 2.** Ablation study of multi-sampling strategy
>
> | Metric                     | Single sampling (copy n times) | Sample n times (n=200) |
> |---------------------------|-----------------|--------------------------|
> | R^2 (without CMA-ES)      | 0.846           | 0.921                    |
> |
>
> **3. CMA-ES improvements:** The high dimensionality of the generative space renders standard CMA-ES inefficient. We introduce two specific modifications in Sec 3.4: Diagonal Covariance Assumption and Top-k Variance Update. These are our SR-specific design that accelerate convergence, prevent the exploration of invalid regions, and reduce equation complexity (supported by ablations in Appendix E.2).
>
> **4. A new Bayesian-inference-based technical route:** To the best of our knowledge, GenSR is the first framework to explicitly formulate SR as a Bayesian inference problem of maximizing p(Equ.|Num.) via ELBO optimization. Training the CVAE corresponds to optimizing the ELBO, while CMA-ES performs the posterior approximation. This provides a novel technical route for SR that goes beyond heuristic search.
> ___

---

> ### Author Response · Authors · 2025-11-22
> **Response to the Reviewer sUgp (part 2)**
>
> * **Advantages of GenSR over contrastive-learning methods  (Q1 & Q2)**
>
> As discussed in lines 51–72 of the revision, the “equation-world map” needed by SR requires two important properties that contrastive learning does not provide or provides insufficiently.
>
> **Advantage 1:  GenSR learns a generative latent space, not a discriminative one.**
> The most important point is contrastive Learning (like SNIP) constructs a discriminative space, which is fragmented and full of "holes." A slight change in a latent vector of discriminative space may decode to an invalid result. Our GenSR constructs a generative space. It learns the distribution of equations —meaning we can sample continuously and decode valid equations. This is a prerequisite for the fine-grained search (like CMA-ES). We add a clearer comparison in Table 3 and Appendix K of revision. Besides, the interpolation experiments (Appendix C.2) confirm that SNIP often falls into semantically invalid areas during search.
>
> &nbsp;&nbsp;&nbsp;&nbsp; **Table 3** comparison between Generative latent space and Discriminative latent space
>
> |                                   | Generative latent space (GenSR) | Discriminative latent space (contrastive) |
> |-----------------------------------|----------------------------------|-------------------------------------------|
> | Learning Objective                | Equation distribution            | Pointwise alignment                       |
> | Continuous latent sampling        | Supported                        | Not supported                             |
> | Suitability for search            | High                             | Low: limited with invalid regions         |
> |
>
> **Advantage 2: GenSR produces a more well-organized latent map.**
> Through symbolic reconstruction and cross-modal distribution alignment, GenSR learns a more structured latent-space than contrastive-learning methods (as analyzed in Sec. 3.3 and visualized in Sec. 5.3). This structure enables efficient coarse localization and fine-grained search. As described in lines 248–249, the symbolic separability of the latent space (guided by symbolic reconstruction) allows GenSR to first localize the search to a high-probability function family, and the local numerical smoothness (enabled by cross-modal distribution alignment) then supports effective fine-tuning. Such latent-space properties have not been considered in prior contrastive-learning-based methods.
>
> ___
> * **Final remarks**
>
> Thank you very much for your thoughtful and constructive review. We hope that our responses have fully addressed your concerns and clarified the contributions of our work. Your feedback has been invaluable in strengthening the manuscript. We sincerely welcome any further suggestions.

---

> ### Author Response · Authors · 2025-11-27
> **Looking forward to discussion**
>
> Dear Reviewer sUgp,
>
> We hope our answers and new experiments have addressed your concerns and questions. Please let us know if you have any more questions before the end of the discussion period.
>
> Thanks for your dedicated service to the community！

---

### Official Review · Reviewer_mtFG · 2025-10-31

**Soundness:** 2
**Presentation:** 2
**Contribution:** 1
**Rating:** 0
**Confidence:** 5

**Summary:**

This paper presents GenSR, a generative SR method that trains a conditional generative model that, at test time (inference/runtime), starts from a good latent-space initialization and refines it with CMA-ES to produce a set of good-fitting equations. The paper claims to introduce a new search paradigm, a Bayesian perspective for SR, and experiments that demonstrate state-of-the-art performance.

**Strengths:**

* The paper is straightforward to read and follow.
* SR is an essential problem for the scientific ML community.
* The interpretability and comparison plots were informative, and a nice visual comparison.

**Weaknesses:**

* Core contributions are not novel, and the authors of GenSR have significant overlap with existing related work that was not discussed at all [1]. [1] proposes a generative SR method using a conditional VAE (CVAE), which encodes expressions into a smooth low-dimensional space, and then applies an evolutionary search algorithm to this latent space, to efficiently sample equations, and at the time of its publication claimed state-of-the-art. Given that the contributions of GenSR are exactly these, I find this missing key related work in direct conflict with this submission. I encourage the authors to: 1) clarify how their method differs, and 2) clearly delineate what is novel and new in their paper in the context of all existing VAE SR methods [1,2] and existing generative methods [3].
  * Using a VAE for SR is not novel and has been extensively done in prior work [1,2], which is conveniently not discussed in this paper. Furthermore, the other missing key related work [2] already shows that using a VAE can improve equation generation under noisy settings, a result also reported in this paper.
* ``GenSR provides a new Bayesian perspective for SR''; this contribution is lacking without appropriate citation and delineation to existing generative SR methods that also provide a Bayesian perspective for SR, such as [3].
* Claim of "State-of-the-art performance" is not validated, as SRBench and corresponding results are old and outdated, dating from 2021/2022. I would encourage the authors to implement the most recent SOTA SR methods and compare them against those to make such a bold claim. Furthermore, the results appear to be cherry-picked, as SR Bench has many problem sets, and only the subset of test equations (Feynman) results are presented graphically in the main figure; whereas other newer SOTA SR methods on SRBench show the same figure across all the equations in the benchmark [5]. I encourage the authors to directly compare to [5] and more recent SR methods to make any claim of state-of-the-art. Furthermore, I encourage the authors to compare against standard problem sets such as Nguyen, Livermore, and similar problem sets as done in [6]. Moreover, the authors omit from the SRBench results the results for "Symbolic Solution (%)" and "Accuracy Solution (%)", leading to suspicion over the results, as these are computed as standard when running the benchmark, and choose to only present R^2 instead. I encourage the authors to be transparent and report all standard results from the benchmark, especially if making any state-of-the-art claims.
* Appendix B seems not necessary, or novel, and appears to be a standard re-statement of the classic textbook ELBO derivation. I would encourage the authors to simply cite this, without the proof in the appendix, as it is not new.

References:
* [1] Mežnar, Sebastian, Sašo Džeroski, and Ljupčo Todorovski. "Efficient generator of mathematical expressions for symbolic regression." Machine Learning 112.11 (2023): 4563-4596.
* [2] Popov, Sergei, et al. "Symbolic expression generation via variational auto-encoder." PeerJ Computer Science 9 (2023): e1241.
* [3] Holt, Samuel, Zhaozhi Qian, and Mihaela van der Schaar. "Deep Generative Symbolic Regression." The Eleventh International Conference on Learning Representations.
* [4] Kamienny, Pierre-Alexandre, et al. "End-to-end symbolic regression with transformers." Advances in Neural Information Processing Systems 35 (2022): 10269-10281.
* [5] Landajuela, Mikel, et al. "A unified framework for deep symbolic regression." Advances in Neural Information Processing Systems 35 (2022): 33985-33998.
* [6] Mundhenk, T. Nathan, et al. "Symbolic regression via neural-guided genetic programming population seeding." arXiv preprint arXiv:2111.00053 (2021).

**Questions:**

* Can the authors clearly delineate how their work is novel, new, and different, compared to the existing highly similar papers such as [1,2,3] stated above, and the broader aspect of VAEs for SR, and similarly other generative models for SR [3]. As it stands with the current submission, this paper does not propose anything novel in context to the existing literature [1,2]; and has severe experimental issues.
* Given that the authors refine constants using BFGS as an inner optimization, have the authors considered the functions getting stuck in local minima and strategies to prevent this, as routinely done in other SOTA SR methods [4]?
* Can you compare experimentally against [4,6,5] by running these competing methods against your method?

---

> ### Author Response · Authors · 2025-11-13
> **Urgent response to Reviewer mtFG regarding the highly sensitive issue.**
>
> Thanks for your feedback. As the concern regarding the alleged “significant overlap” and “highly similar” are particularly sensitive, we address this review first. Other comments will be responded to shortly.
> ***
> * **Factual Mistake of Reviewer mtFG on [1]**
>
> The Reviewer mtFG claims that our work “has significant overlap” with [1], asserting that “[1] proposes a generative SR method using a conditional VAE (CVAE)…”, which is a **severe factual mistake**.
>  **[1] does not use a Conditional VAE, and the word “conditional” does not appear anywhere in this paper. In fact, [1] employs a Hierarchical VAE (HVAE). We have confirmed with the original authors of [1] via email that the “CVAE” mentioned in [1] is not an abbreviation for Conditional VAE, but refers to a baseline method named Chemical VAE.**
> This can be easily verified from the citations in [1] and official GitHub repository (https://github.com/aspuru-guzik-group/chemical_vae).
>
> We respectfully urge both reviewers and the AC to uphold academic rigor and fairness, particularly when addressing sensitive issues of research overlap.
>  ***
>
> * **Differences from [1,2,3]**
>
> Thank you for your careful reading. We acknowledge our oversight in not thoroughly reviewing the journal papers. This feedback is very valuable to us, and we will include a discussion and citations in the revised version. However, we **strongly disagree** with the assertion that our work is “highly similar to [1,2,3]”.
>
> **1. Core Contribution:** We emphasize that our core contribution lies in **constructing a well-organized generative latent space that ensures both symbolic continuity and local numerical smoothness—properties that are absent and unattainable in [1,2,3]—rather than simply applying a VAE to build a latent space.**
> As stated in Lines 60–68 of our paper: Methods such as [1,2], which use VAE solely to reconstruct symbolic equations, cannot leverage numerical errors as effective guidance in the latent space. Besides, purely numerical latent spaces (e.g., [3] and other end-to-end models) lead to unstable decoders. To address this, we designed a dual-branch Conditional VAE framework that leverages bimodal input to jointly model symbolic and numerical continuity in the latent space, which unimodal designs like [1,2,3] cannot achieve. Our dual-branch CVAE is not a standard CVAE framework but a customized design driven by our specific motivation, as Reviewer A4iV noted, “ This design is clearly motivated”.
>
> **2. Probabilistic Perspective:**
> We do not claim to be the first to view SR from a Bayesian perspective; rather, we analyze the probabilistic rationale of GenSR. Nevertheless, our probabilistic formulation fundamentally differs from [3]. **GenSR employs an explicit variational Bayesian model that directly estimates the conditional posterior p(Equ.|Num.), strictly following the ELBO optimization principle.** In contrast, **[3] only approximates p(Token|Num.) at the token level without constructing a latent distribution over symbolic equations**. Moreover, in comparison with [1,2], whose inputs are unimodal (symbolic equation only), their objectives maximize p(Equ.) rather than the conditional probability p(Equ.|Num.). Since SR inherently seeks the most probable equation given the observed numerical data, our method aligns directly with the true probabilistic goal of the SR task, whereas [1,2] do not.
>  ***
>
> * **Additional Clarifications:** We have never claimed to be the first to use VAE for SR or to build a generative latent space for equations. The Conditional VAE in our framework is only an implementation tool; our focus is on analyzing and realizing effective latent-space properties.
> ***
>
> We appreciate the Reviewer mtFG’s feedback and will revise our manuscript to include the citations and clarifications. **However, if the Reviewer mtFG’s score of 0/10 and confidence level of 5/5 were based on obvious factual inaccuracies and a misunderstanding of our contributions, we regretfully find this unacceptable.** We respectfully request the Reviewer mtFG to reconsider both the score and the confidence level.
>
>
> [1] Mežnar, Sebastian, Sašo Džeroski, and Ljupčo Todorovski. "Efficient generator of mathematical expressions for symbolic regression." Machine Learning 112.11 (2023): 4563-4596.
> [2] Popov, Sergei, et al. "Symbolic expression generation via variational auto-encoder." PeerJ Computer Science 9 (2023): e1241.
> [3] Holt, Samuel, Zhaozhi Qian, and Mihaela van der Schaar. "Deep Generative Symbolic Regression." The Eleventh International Conference on Learning Representations.

---

> ### Author Response · Authors · 2025-11-22
> **Response to the Reviewer mtFG’s remaining comments (part 1)**
>
> We have already responded Weaknesses 1 and 2 as well as Question 1 in our previous response, and added the corresponding citations and clarifications in the revised version (line 60-64, line 307-314). Below, we address the remaining comments from the reviewer.
> ___
>
> * **On the claim that SRBench is “old and outdated” (W3)**
>
> Reviewer mtFG argues that SRBench (2021) is “old and outdated”, and suggests using “standard problem sets such as Nguyen (2011), Livermore(2021)”. However, it is an inaccurate statement and suggestion, Because:
>
> 1.	SRBench is newer, richer, and more challenging than the benchmarks (Nguyen (2011) and Livermore (2021)) which are suggested by reviewer. A comparison is shown in Table 1.
>
> 2.	SRBench has been the primary and standard benchmark for recent SR research (MDLFormer (ICLR 2025), RAG-SR(ICLR 2025), SNIP(ICLR 2024), TPSR(NIPS 2023), etc., all use SRBench). As stated in the newly released LLM-SRBench (ICML 2025 Oral): “The scientific equation discovery benchmarks are primarily represented by SRBench.” Moreover, among the reviewer-cited works [1-5], three use SRBench (while [6] was released the same year as SRBench).
>
> &nbsp;&nbsp;&nbsp;&nbsp;**Table 1** comparison of the three benchmarks
> |                      | Nguyen | Livermore | SRBench (we used) |
> |----------------------|--------|-----------|--------------------|
> | Release Year         | 2011   | 2021      | 2021               |
> | Total Number of Problems | 12 | 22        | 252                |
> | Maximum Number of Variables | 2 | 2       | 10                 |
> |
>
> Thus, we respectfully **DISAGREE** that our benchmark is “old and outdated”. We will nevertheless add results on Nguyen and Livermore in Table 2.
>
> ___
>
> * **On the claim that our baselines are outdated (W3)**
>
> Reviewer mtFG encourage us to implement and compare the most recent SOTA SR methods, like [4,5,6] (2021-2022). However, it is an inaccurate suggestion, Because:
>
> 1. We include 18 baselines, covering two ICLR 2025 works (MDLformer, RAG-SR), two ICLR 2024 works (RSRM, SNIP). **All of these are NEWER than ALL reviewer suggested and cited work [1-6] (2021-2023)**.
> Therefore, we respectfully **DISAGREE** with the statement that our baselines are not new enough. Nevertheless, we add comparisons with [5,6] in Table 2.
>
> 2. Notably, the reviewer-suggested baseline [4] is already one of our key comparison methods. All our tables and figures include results for [4]. We kindly ask the reviewer to reconsider this comment after a careful reading.
>
> ___
>
> * **On the accusation of “cherry-picking results” (W3)**
>
> We did **NOT** only show the Feynman subset in our paper. The main text visualizes both Feynman (Fig.2) and Strogatz datasets (Fig.3). In Sec. 5.1 (line 327), we clearly state that Appendix G.5 contains the complete numerical results for Feynman, Strogatz, and Black-box datasets—including accuracy, time complexity, and equation complexity. Undoubtedly, precise numerical comparisons are more detailed and more convincing than visual plots (like Pareto front results). Due to page limits, most conference papers only show partial results in the main text.
>
> Thus, we strongly **DISAGREE** with the accusation of “cherry-picking”, which stems from overlooking the appendix.
>
> ___
>
> * **On the request for more evaluation metrics (W3)**
>
> Thank you for the suggestion. We must clarify that we did not “choose to only present $R^2$”.Besides $R^2$ , we report equation complexity and time complexity, which are equally important in SR (corresponding to interpretability and efficiency). Our method aims to a balanced trade-off across all three targets, so we avoid over-emphasizing any single metric. However, we are happy to include “Symbolic Solution (%)” and “Accuracy Solution (%)” for additional accuracy comparisons, please refer to Table 2.
>
> ___
>
> &nbsp;&nbsp;&nbsp;&nbsp;**Table 2** More results on Nguyen and Livermore
>
> | Method   | Nguyen Symbolic Solution | Nguyen Accuracy Solution | Livermore Symbolic Solution | Livermore Accuracy Solution |
> |----------|------------------|------------------|----------------------|----------------------|
> | HVAE[1]  | 55.3             | 83.8             | 28.2                 | 44.7                 |
> | SEGVAE[2]| 47.5             | 86.7             | 21.8                 | 49.3                 |
> | uDSR[5]   | 75.4             | 92.8             | 61.3                 | 74.2                 |
> | NG-GP[6]   | 61.5             | 89.4             | 53.7                 | 78.5                 |
> | E2E[4]     | 34.9             | 84.9             | 15.8                 | 36.8                 |
> | GenSR (ours) | 73.1             | 95.3             | 63.4                 | 82.2                 |
> |
> ___

---

> ### Author Response · Authors · 2025-11-22
> **Response to the Reviewer mtFG's remaining comments (part 2)**
>
> * **On our claim of “State-of-the-art performance” (W3)**
>
> We thank the reviewer for raising this point. Our claim does not refer to being the best on a single metric. Instead, as stated in lines 83-84, our method lies on the Pareto frontier across three targets: accuracy, equation complexity, and efficiency. To our knowledge, no previous method achieves optimality on all three targets simultaneously. We acknowledge that the statement may cause ambiguity and have revised the expression in lines 90–91.
>
> ___
>
> * **On the use of BFGS (Q2)**
>
> We use the BFGS to refine the function’s constants same as [4], which is common practice in other SR methods (such as SNIP).
>
> ___
>
> * **On Appendix B (Q3)**
>
> We thank the suggestion. However, we emphasize that, to our knowledge, GenSR is the first framework to perform SR by estimating and optimizing the ELBO of p(Equ.|Num.). Thus, presenting the derivation in the appendix rather than the main paper is a reasonable choice.
>
> ___
>
> * **Final remarks**
>
> We appreciate the reviewer’s time. However, **the key issue—“outdated baselines”, “outdated benchmark”, and “only showing the Feynman subset”—are inconsistent with the facts**. We sincerely encourage the reviewer to reconsider the statements such as “severe experimental issues” and “cherry-picked results”, and to reconsider the score and confidence with a more careful reading.
>
> ___
> [1] Mežnar, Sebastian, Sašo Džeroski, and Ljupčo Todorovski. "Efficient generator of mathematical expressions for symbolic regression." Machine Learning 112.11 (2023): 4563-4596.
> [2] Popov, Sergei, et al. "Symbolic expression generation via variational auto-encoder." PeerJ Computer Science 9 (2023): e1241.
> [3] Holt, Samuel, Zhaozhi Qian, and Mihaela van der Schaar. "Deep Generative Symbolic Regression." The Eleventh International Conference on Learning Representations.
> [4] Kamienny, Pierre-Alexandre, et al. "End-to-end symbolic regression with transformers." Advances in Neural Information Processing Systems 35 (2022): 10269-10281.
> [5] Landajuela, Mikel, et al. "A unified framework for deep symbolic regression." Advances in Neural Information Processing Systems 35 (2022): 33985-33998.
> [6] Mundhenk, T. Nathan, et al. "Symbolic regression via neural-guided genetic programming population seeding." arXiv preprint arXiv:2111.00053 (2021).

---

> ### Author Response · Authors · 2025-11-27
> **Looking forward to discussion**
>
> Dear Reviewer mtFG
>
> We hope our answers and new experiments have addressed your concerns and questions. Please let us know if you have any more questions before the end of the discussion period.
>
> Thanks for your dedicated service to the community！

---

### Author Response · Authors · 2025-11-23
**Main updates of the revision**

We would like to thank Reviewer sUgp, CAH6, and A4iV for their constructive feedback and appreciate their positive comments. We have revised the manuscript, and all changes are highlighted in blue. The main updates are as follows:

1.	We added the references [1, 2, 3]  and clarified the advantages of GenSR over these methods (lines 60–64, 306–314) (suggested by mtFG).

2.	We included a comparison with CMA-ES and BO in Appendix H (suggested by A4iV).

3.	We added an ablation study of the dual-branch CVAE framework in Appendix I (suggested by CAH6).

4.	We provided additional clarification on the behavior of the KL-divergence term in Appendix J (suggested by A4iV).

5.	We supplemented a comparison between the discriminative and generative latent spaces in Appendix K (suggested by sUgp).

___

[1] Mežnar, Sebastian, Sašo Džeroski, and Ljupčo Todorovski. "Efficient generator of mathematical expressions for symbolic regression." Machine Learning 112.11 (2023): 4563-4596.
[2] Popov, Sergei, et al. "Symbolic expression generation via variational auto-encoder." PeerJ Computer Science 9 (2023): e1241.
[3] Holt, Samuel, Zhaozhi Qian, and Mihaela van der Schaar. "Deep Generative Symbolic Regression." The Eleventh International Conference on Learning Representations.

---

### Author Response · Authors · 2025-12-04
**Summary of Discussion Results**

During the discussion period, we did our best to address all reviewers’ questions and concerns. Reviewer **A4iV** indicated a willingness in the initial review to raise the score 6 and confidence 3 upon clarification. After reviewing our detailed response, reviewer **A4iV** was willing to increase both the confidence and the score (**from 6 to 8**), and expressed satisfaction with the quality of our paper and our response. Reviewers **sUgp** and **CAH6** did not respond during the discussion, but they had provided positive scores in the first round, with scores of **6 (confidence 3)** and **8 (confidence 5)**, respectively.

We appreciate A4iV’s strong support for our work and for noting that “***Although one reviewer strongly opposed the paper with the highest confidence score before the deadline, I still believe that the contributions of this paper are different from prior work.***” We also appreciate reviewers sUgp and CAH6 for their positive score and comments, including recognizing **our comprehensive experiments, well-written manuscript, clear comparisons, and clear theoretical framework**.

However, we **strongly disagree** with reviewer **mtFG’s multiple false claims**, including misinterpretations of cited works, incorrect statements about our experiments, and serious neglect of our work. More importantly, reviewer mtFG made an extremely sensitive allegation of “significant overlap with [1]” seems without even reading paper [1]. These inaccuracies directly resulted in an excessively negative score  (**0/10 with 5/5 confidence**).  On the day after the scores were released, we submitted an urgent response to point out the reviewer mtFG’s serious false allegations, unfortunately, we received no further response throughout the entire discussion period.

At last, we respectfully urge all members of the community to uphold academic rigor and fairness.

---

### Meta-Review · Area_Chair_bGJj · 2026-01-07

**Summary:**

This paper proposes a method to construct a latent space for symbolic regression, considering the numerical properties of equations. While three reviewers support acceptance, one reviewer raised several concerns about missing related works, baselines, evaluation metrics, and performance. The reviewer did not respond to the authors' response; however, I found that the authors properly addressed all concerns of the negative reviewer. Hence, I recommend the acceptance of this submission.

**Reviewer Concerns:**

There are some (possibly) unsolved concerns raised by reviewers. Most importantly, Reviewer mtFG raised several concerns about missing related works, baselines, evaluation metrics, and the performance of the proposed method. Since the reviewer did not respond to the authors' response during the discussion period (and the discussion period accidentally terminated), one could not clearly expect the final thought of this reviewer after reading the authors' response. However, I think the authors properly addressed the proposed concerns given that the authors (1) clearly state the difference to the suggested related work, (2) evaluate metrics other than $R^2$, and (3) include experimental results with suggested baselines. While the proposed method does not achieve the state-of-the-art performance, the authors clarified that their method is empirically Pareto-optimal.

Reviewer A4iV also raised a concern about the optimization in the proposed latent space. This concern has not been solved in the authors' response; however, I believe the current contribution is enough for the publication, and it may be out of the scope of this work.

**Reviewer Scores:**

Reviewer A4iV would increase their score from 6 to 8 as they indicated in the comments. After reading the authors' response, I think Reviewer mtFG may increase the score. However, I do not expect the reviewer to support the acceptance based on their first review. I expect that the other two reviewers would not decrease their scores.

---

### Decision · Program_Chairs · 2026-01-26

Accept (Poster)